# Phylogenomics uncovers early hybridization and adaptive loci shaping the radiation of Lake Tanganyika cichlid fishes

Iker Irisarri [1,8], Pooja Singh [1,2], Stephan Koblmüller [2], Julián Torres-Dowdall [1], Frederico Henning[1,3], Paolo Franchini [1], Christoph Fischer[4,5], Alan R. Lemmon [6], Emily Moriarty Lemmon [7], Gerhard G. Thallinger [4,5], Christian Sturmbauer[2] & Axel Meyer [1,9]

Lake Tanganyika is the oldest and phenotypically most diverse of the three East African cichlid fish adaptive radiations. It is also the cradle for the younger parallel haplochromine cichlid radiations in Lakes Malawi and Victoria. Despite its evolutionary significance, the relationships among the main Lake Tanganyika lineages remained unresolved, as did the general timescale of cichlid evolution. Here, we disentangle the deep phylogenetic structure of the Lake Tanganyika radiation using anchored phylogenomics and uncover hybridization at its base, as well as early in the haplochromine radiation. This suggests that hybridization might have facilitated these speciation bursts. Time-calibrated trees support that the radiation of Tanganyika cichlids coincided with lake formation and that Gondwanan vicariance concurred with the earliest splits in the cichlid family tree. Genes linked to key innovations show signals of introgression or positive selection following colonization of lake habitats and species' dietary adaptations are revealed as major drivers of colour vision evolution. These findings shed light onto the processes shaping the evolution of adaptive radiations.

[1] Lehrstuhl für Zoologie und Evolutionsbiologie, Department of Biology, University of Konstanz, Universitätsstrasse 10, Konstanz 78457, Germany. [2] Institute of Biology, University of Graz, Universitätsplatz 2, Graz 8010, Austria. [3] Department of Genetics, Institute of Biology, Federal University of Rio de Janeiro, Ilha do Fundão, Rio de Janeiro 21944-970, Brazil. [4] Institute of Computational Biotechnology, Graz University of Technology, Petersgasse 14, Graz 8010, Austria. [5] OMICS Center Graz, BioTechMed Graz, Stiftingtalstraße 24, Graz 8010, Austria. [6] Department of Scientific Computing, Florida State University, Dirac Science Library, Tallahassee, FL 32306, USA. [7] Department of Biological Science, Florida State University, Biomedical Research Facility, Tallahassee, FL 32306, USA. [8] Present address: Department of Biodiversity and Evolutionary Biology, Museo Nacional de Ciencias Naturales (MNCN-CSIC), José Gutiérrez Abascal, 2, Madrid 28006, Spain. [9] Present address: Radcliffe Institute for Advanced Study, Harvard University, Cambridge 02138 MA, USA. These authors contributed equally: Iker Irisarri, Pooja Singh.  Correspondence and requests for materials should be addressed to C.S. (email: christian.sturmbauer@uni-graz.at) or to A.M. (email: axel.meyer@uni-konstanz.de)

Rapid speciation, when coupled with ecological opportunity, can give rise to a multitude of species within short time periods, a process known as adaptive radiation[1]. Several lineages have diversified through this process, including Hawaiian silverswords[2], Darwin's finches[3], and East African cichlids[4]. Investigating adaptive radiations is crucial for better understanding the factors that drive speciation[5]. However, disentangling the geographical setting, ecology, genetic causes, commonalities and peculiarities in different adaptive radiations has proven difficult[6,7]. Sudden abiotic changes such as new volcanic islands or lakes can provide empty niches that facilitate diversification. Speciation can also be promoted by key traits that allow the exploitation of new niches, or gene flow between divergent lineages that produce transgressive phenotypes showing faster responses to divergent selection[8].

Cichlid fishes (Cichlidae), with >1700 described species, are a model system for the study of adaptive radiation, ecomorphological diversification, and speciation[4,6]. East African Great Lakes harbor ~90% of all the described cichlid species so far[9], which are the result of a successful evolution shaped by vicariance-driven and opportunity-driven speciation[10]. Understanding the evolutionary history of cichlids and the factors contributing to their diversification have been major quests for an ever-growing community of evolutionary biologists. However, despite decades of research, some crucial evolutionary relationships remain unresolved and also the timing of divergences among the major lineages is still controversial. Thus, establishing a robust time-calibrated phylogeny is a first requirement for tracing the patterns of diversification (in time and space) and to better understanding the factors that shaped their adaptive radiation. In addition, the role and relative importance of hybridization, ecology, or life history in shaping cichlid radiations remains poorly understood.

Phylogenetic inference of adaptively radiated lineages has proven difficult both with morphology[4] and molecules[11]. Fast diversification prevents the accumulation of shared derived changes and increases the retention of incomplete lineage sorting (ILS) in the genomes, whereas ecomorphological convergence and hybridization further confound phylogenetic inference methods. Despite the many studies that have contributed to clarifying the phylogeny of cichlids (reviewed in Koblmüller et al.[12]), considerable disagreements still exist regarding key relationships, partly due to the few and lowly informative markers available and the scarcity of fossils, which are also problematic due to frequent parallel evolution of similar ecomorphologies[13,14]. One of the main open questions is the deep phylogenetic structure of the Lake Tanganyika radiation, the oldest among East African Great Lakes. Lake Tanganyika hosts the ecologically, morphologically, and genetically most diverse species assemblage[12]. It also holds the key to understanding the origin of haplochromine cichlids, the most species-rich lineage that seeded the radiations in Lakes Malawi and Victoria and also re-colonized Lake Tanganyika[9,15]. While the monophyly of cichlid tribes is well established, their interrelationships remain hotly debated[11,15–18]. Several studies have investigated cichlid macroevolution[19], but given current phylogenetic uncertainties, employed trees might not reflect the 'true' species history[16]. A robust phylogenetic hypothesis for cichlids will not only resolve existing controversies but it will also allow to understand the origin and evolution of key innovations, life history changes, and further clarify the patterns of convergence and parallel evolution that characterize cichlid radiations.

The rapid diversification of East African cichlids has been attributed to key innovations including (i) functional decoupling of oral and pharyngeal jaws facilitating exploitation of diverse trophic sources[20], (ii) vision adaptation to different water turbidity[21], and (iii) body coloration associated with sexual selection

and reproductive isolation[22]. The relative importance of these innovations is poorly understood, given that most previous studies have focused on the effect of single factors on these traits, or analysed traits in isolation. The lack of studies simultaneously modelling different ecological and life-history factors hampers the identification of hierarchical relationships among them, as well as the identification of lineage-specific shifts[6].

In addition to key innovations, hybridization has also been proposed as a major factor shaping East African cichlid radiations[8]. Hybridization has been demonstrated at the onset of the Lake Malawi and Victoria cichlid flocks[23,24] and suggested to fuel these radiations. Some studies proposed a similar scenario for Lake Tanganyika[18,25], but so far no study has demonstrated the presence of hybridization at the base of this radiation, an imperative for invoking its role in boosting this diversification.

The integration of molecular dating with geological events can identify relevant ecological factors facilitating radiations and provide an independent control for hybridization hypotheses. The Lake Tanganyika species flock is assumed to have originated in step with the lake formation and maturation 12–9 Ma[26]. However, molecular dating analyses have suggested markedly different ages from ca. 51 to 16 Ma[9,27–31]. Controversy persists also regarding the age estimates for the deepest divergences in the cichlid family tree. These were initially assumed to follow Gondwanan vicariance[32–35]. Recent molecular dating recovered significantly younger ages for these splits that contradict the vicariance scenario and therefore hypothesized the long-distance trans-oceanic dispersal of cichlids[25,29,30]. In contrast, earlier molecular datings using vicariant calibrations estimated much older divergences in African cichlids that are difficult to reconcile with the age and geological history of the African Great Lakes and the biology of their endemic species[9,12]. The difficulty in estimating consistent divergences simultaneously at deeper and shallower regions of the tree originates in part from the scarcity of reliable fossils for calibration (the oldest known is only 45 Myr old[36]) and the lack of a reliable molecular clock rate estimate for cichlids.

Here, we attempt to reconstruct a robust cichlid family tree focusing on the Lake Tanganyika flock, which is time-calibrated employing a newly described East African cichlid fossil[37]. Phylogenetic networks and Patterson's *D*-statistics are used to examine hybridization signals among cichlid tribes. We find evidence for hybridizations at the bases of the Lake Tanganyika flock and the tribe Haplochromini that seeded the radiations in Lakes Malawi and Victoria. These findings support the role of hybridization in boosting all major cichlid radiations in East Africa. Using the new phylogenetic framework, we analyse the molecular evolution of genes associated with key cichlid innovations. We find positive selection associated with the colonization of Lake Tanganyika and identify diet as one of the most important ecological factors.

## Results and Discussion

**Anchored phylogenomics resolves Lake Tanganyika radiation.** Our dataset comprises 533 anchored loci for 149 species (950,518 filtered aligned nucleotide positions, 5.1% missing data), representing all major cichlid lineages with emphasis on the Lake Tanganyika radiation (Supplementary Table 1). Both the coalescent (Fig. 1) and the concatenated tree (Supplementary Fig. 1) produced highly (~91%) congruent tree topologies with most nodes receiving full bootstrap support. Differences between both trees (27/297 or ~9% of nodes) are restricted to East African cichlids and mostly affected relationships within tribes, with two exceptions (see below). The difficulty of reliably reconstructing inter-tribal relationships in Lake Tanganyika is well illustrated by

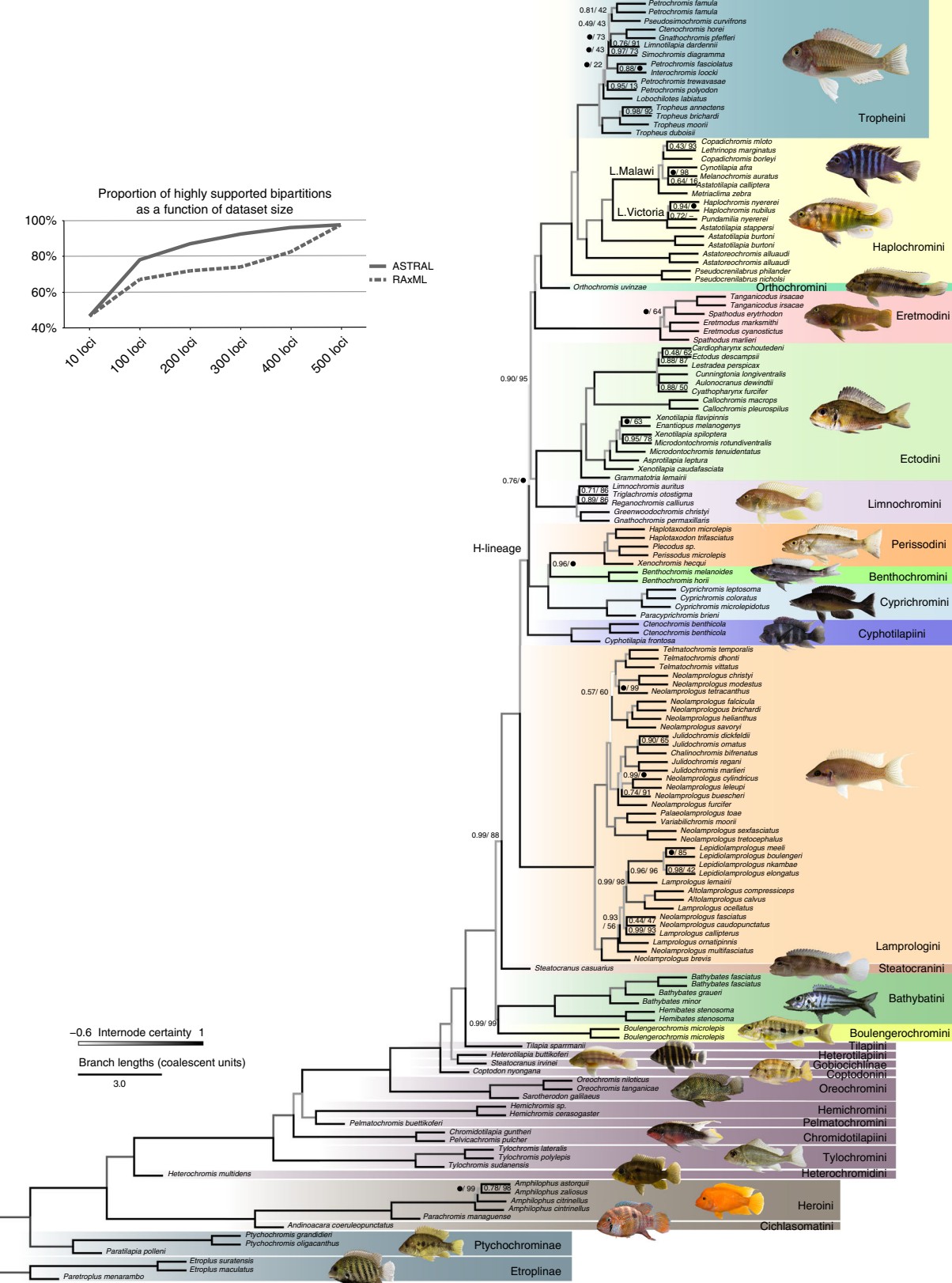

**Fig. 1** Coalescent species tree of cichlids (ASTRAL). Numbers at nodes are support values from local posterior probabilities and multi-locus bootstrapping proportions, respectively, and black dots represent full support. Nodes without actual numbers received full support from both measures. Branch lengths are in coalescent units and branch colours reflect internode certainty values. The inset graph shows the percentage of highly-supported nodes (>75% locus jackknife proportion) reconstructed by ASTRAL and RAxML with increasing number of loci. Photo credits: Wolfgang Gessl

the low internode certainty[38] for these nodes, in contrast to the relatively high values (>80%) for the monophyly of most tribes (Fig. 1 and Supplementary Fig. 2). The congruence between coalescent and concatenated trees might be surprising at first, given that concatenation, unlike coalescence, does not account for ILS. In fact, the low internode certainty derives mostly from short branches, which supports the presence of ILS[39] (Supplementary Fig. 3). Gene jackknife analyses show that ILS negatively impacts concatenation (RAxML) using 400 loci or less whereas coalescence (ASTRAL) consistently recovers more final-tree bipartitions (Fig. 1). Using 500 or more loci instead, both methods converge to very similar topologies, suggesting that the genuine phylogenetic signal can overcome the confounding effect of ILS ignored by concatenation. In addition, using more genes probably dilutes the adverse effect of gene flow events (e.g., introgression) on both concatenation and coalescence approaches.

The monophyly of all cichlid tribes was unambiguously recovered (Fig. 1). The monophyly of the East African cichlid radiation[11,15,17,40,41] is confirmed after the inclusion, for the first time, of several riverine species currently distributed outside of Lake Tanganyika that are essential to test this monophyly. The deepest splits in the tree correspond very well with distribution patterns across Gondwana-derived landmasses: successive branching of Etroplinae (Madagascar and India/Sri Lanka), Ptychochrominae (Madagascar), and Neotropical (Cichlasomatini + Heroini) and African cichlids as sister clades[28,31]. As in the most recent and taxonomically comprehensive study of African cichlids[41], we recovered early-branching of Heterochromidini and Tylochromini but found alternative relationships among most other tribes (Fig. 1). In particular, we recovered strong support for Tilapiini as sister to a clade composed of Steatocranini and the Lake Tanganyika species flock.

The coalescent species tree recovered Steatocranini (*Steatocranus casuarius*) within the Lake Tanganyika flock, as sister to all other tribes to the exclusion of Bathybathini + Boulengerochromini (Fig. 1), whereas the maximum likelihood tree (Supplementary Fig. 1) recovered Steatocranini as sister group to the Lake Tanganyika species flock that includes Bathybathini and Boulengerochromini. We interpret this difference to result from hybridization (see below). The second major inter-tribal topological disagreement pertained to relationships within the so-called H-lineage[42] (see also Fig. 1): coalescent analyses recovered the successive branching of (i) Cyprichromini as sister group to Benthochromini + Perissodini and (ii) Limnochromini + Ectodini, whereas concatenated analyses recovered these two clades as a sister group. Previous studies generally agree with our recovered position for Haplochromini (including Tropheini) and the deeper branching of Boulengerochromini, Bathybatini, and Lamprologini, but the precise inter-relationships among all other tribes differed significantly[11,15,17,18,25,41]. The Eretmodini, whose placement has been controversial in previous studies[14,43–45] is found within the H-lineage[11,17,25,30,42], specifically as the sister group of Orthochromini + Haplochromini (including Tropheini). Within the Haplochromini, the riverine genera *Astatoreochromis* and *Astatotilapia* branch off prior to the two major clades representing the adaptive radiations in Lake Victoria (>500 spp.) and Lake Malawi (>800 spp.). Our phylogeny supports the evolution of Tanganyika-endemic Tropheini from riverine ancestors that re-colonized Lake Tanganyika and diversified in parallel with the already existing radiation[11,12,15].

**Cichlid timetree reconciles two major geological events**. We performed independent timetree analyses using 10 calibration schemes based on fossil or biogeography (Supplementary Table 2). These analyses provided a comprehensive understanding of the effect of fossil- or vicariance-based calibrations on both the deep and recent parts of the cichlid tree. Notably, this is the first time the Lake Tanganyika radiation is anchored with the recently described fossil †*Tugenchromis pickfordi*[37], and we evaluate its effect by comparing calibration schemes with or without this fossil. Our divergence time estimates dated the origin of the Lake Tanganyika species flock to the late Miocene, approximately 13.7–12.7 Ma for calibration schemes C06-C10 (Fig. 2) and slightly more recently (11.1–5.9 Ma) for calibration schemes C01-C05 not including †*T. pickfordi* (Supplementary Data 1). Taking into account 95% confidence intervals, both ranges are congruent with a diversification of the Lake Tanganyika species flock associated with the colonization of the new lake basin (originated 12–9 Ma[46]), which provided new niches and ecological opportunities required for adaptive radiation[10]. Likewise, it is not possible to exclude an alternative scenario of an earlier origin of some major lineages. However, the fact that all Tanganyika-endemic tribes, except the early-branching Lamprologini, occupy distinct gross fundamental niches (ecological guilds)[12] suggests that the initial radiation likely occurred within the confines of the emerging lake system. Instead, if much of the initial divergence had happened in different rivers systems prior to the lake formation (i.e., in allopatry) one could expect each lineage to radiate into an array of species occupying broader niches, which contradicts the pattern observed today. Importantly, members from cichlid tribes of the Lake Tanganyika flock are not present in any nearby river system, with only three exceptions: (i) Orthochromini, found exclusively in the Malagarazi drainage and nearby rivers that drain into Lake Tanganyika; (ii) the Haplochromini that originated in Tanganyika, colonized large parts of Africa, and later re-colonized Tanganyika too (Tropheini); and (iii) the Lamprologini that contain two lineages that dispersed from Lake Tanganyika into the Congo and Malagarazi river systems[47]. Again, an origin of Lake Tanganyika species flock much earlier than the lake formation would need to explain the extinction of many riverine members of Tanganyikan tribes. In fact, other closely related cichlid lineages, some of which have also colonized Lake Tanganyika[41], continue to occur in nearby water bodies.

Our analyses date the origin of the cichlid family to between the Jurassic and Early Cretaceous, depending on the assumed calibration scheme, clearly predating the oldest known cichlid fossil (~46 Ma[36]). The estimated divergence times for the deepest splits in the tree, namely those between Etroplinae-Ptychochrominae at 175.8–121.0 Ma and between African and Neotropical cichlids at 143.9–98.9 Ma, are consistent overall with a vicariance scenario of Gondwanan fragmentation[27,28,34,35] but contradict other proposed scenarios[29–31]. The Gondwanan vicariance is robustly supported by all fossil-based calibration schemes, with the exception of schemes C03 and C08, which produce seemingly biased older estimates and larger confidence intervals (Supplementary Data 1). The difference between schemes C03 and C08 with their analogous C02 and C07 is the placement of †*Mahengechromis*, which has an uncertain phylogenetic position[48]. Its position is likely closer to the origin of all African cichlids rather than the node immediately after (excluding *Heterochromis*) and thus it should be applied to calibrate this event. The use of maximum bounds in schemes C05 and C10 diminishes the detrimental effect of misplacing †*Mahengechromis* (Supplementary Data 1). The vicariance scenario proposed here more easily explains current distribution patterns of cichlids across Gondwanan-derived landmasses without requiring trans-oceanic dispersal to have occurred. The latter would require strong physiological tolerance for salinity conditions in which most modern cichlids would not survive (apart from a few species that occur in, or can tolerate brackish or marine conditions[49,50]).

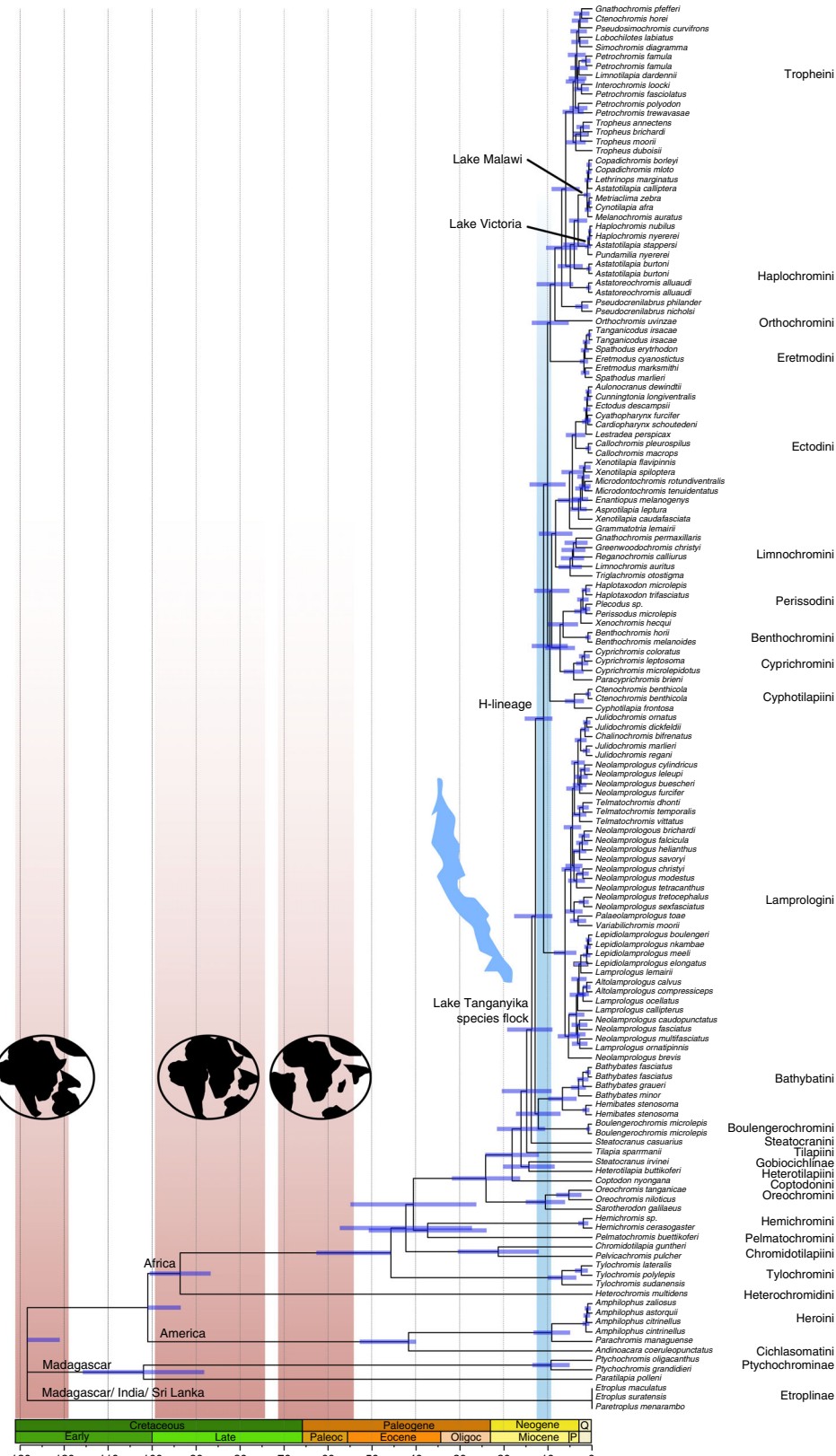

**Fig. 2** Time-calibrated phylogeny of cichlids. Divergence times have been inferred with RelTime with calibration scheme C10. Node bars are 95% confidence intervals. Vertical shadows represent splits of Gondwanan-derived landmasses (brown) and the formation of Lake Tanganyika (blue). Relevant (current) distribution patterns of main cichlid lineages and the colonization events of East African Great Lakes are indicated. Scale is in million years and main geological periods are highlighted. Detailed divergence times for all individual nodes and timetrees under calibration schemes C01–C10 are available in Supplementary Data 1

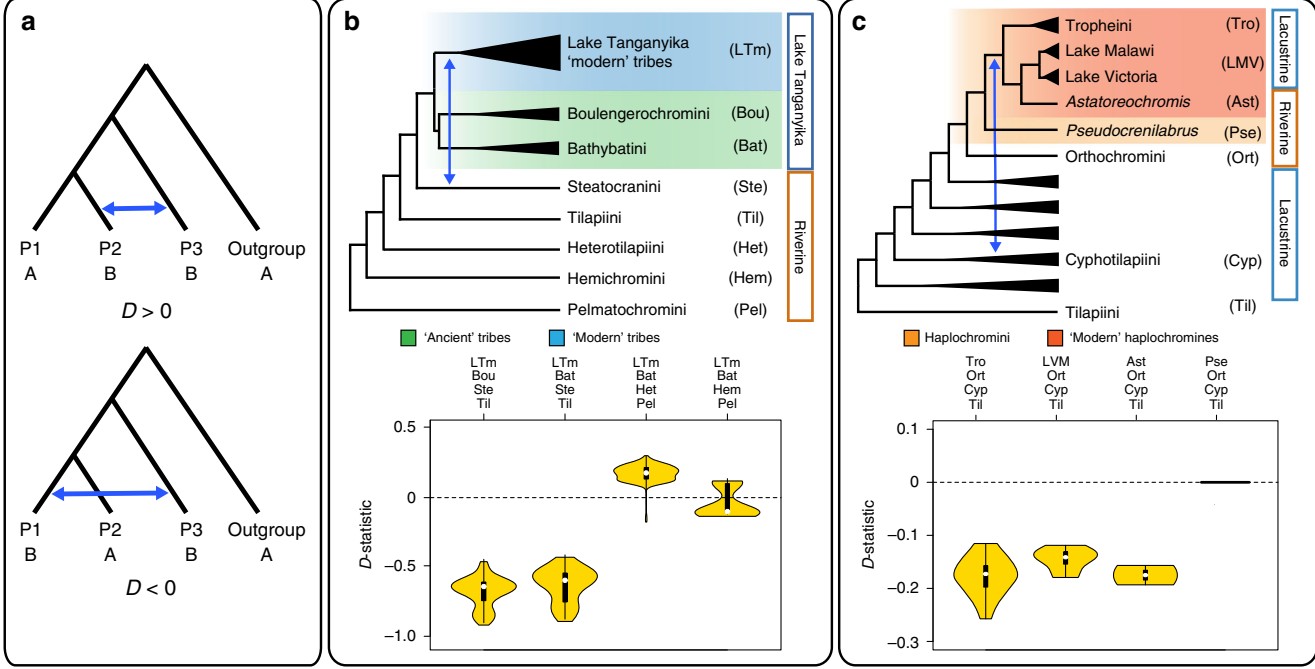

**Fig. 3** Hybridization in the Lake Tanganyika species flock. **a** Schematic representation of Patterson's $D$ test. Blue arrows represent gene flow between distantly related lineages, either P2–P3 (ABBA excess; $D > 0$) or P1–P3 (BABA excess; $D < 0$). **b** Proposed scenario for the hybridization between Steatocranini and 'modern' tribes of Lake Tanganyika. **c** Hybridization between Cyphotilapiini and haplochromines. Violin plots show the distributions of Patterson's $D$-statistics from individual-based permutations using different taxa combinations, as labeled in the phylogeny. Taxa configurations follow the concatenated maximum likelihood tree (Supplementary Fig. 1) because it allows testing the hypothesized hybridization events

In addition, the estimated divergence of the Tanganyika species flock agrees with the radiation occurring within the confines of the incipient lake.

The recent study proposing long-distance trans-oceanic dispersal of cichlids inferred divergences that are seemingly too old (~50 Ma) for the Tanganyika species flock[31]. In addition to substantial differences in dating methodology, the disagreements in divergence times between this and our analyses could be partly due to the underlying molecular dataset, which was substantially smaller (40 loci) and dominated by mitochondrial markers (~66%) and missing data (~40%). Expanding our anchored loci dataset to include other non-cichlid taxa, which is straightforward for anchored loci, would allow the incorporation of additional informative fossils in the outgroup as in Matschiner et al.[31] and help clarify current disagreements on cichlid diversification. Further insight into the apparent contradiction between existing cichlid timetrees would be possible by conducting an in-depth study comparing different dating software/methodologies and molecular datasets with comparable assumptions and models.

**Ancient hybridizations fueled East African radiations**. We investigated our dataset for signals of hybridization/introgression by analysing high-frequency discordances in galled networks[51], and propose hypotheses of inter-tribal hybridization that are later tested using Patterson's $D$ statistic (or ABBA/BABA test)[52]. Galled networks show substantial topological discordance among loci (Supplementary Fig. 4) which can originate from either ILS or hybridization. However, given the stochasticity of ILS, reticulate events supported by a relatively high number of gene trees likely represent hybridization events. Following this strategy, we hypothesize two inter-tribal hybridizations: (i) a first one between Steatocranini (*Steatocranus casuarius*), the ancestor of Bathybathini + Boulengerochromini, and the remaining lineages ('modern' tribes) of the Lake Tanganyika flock, and (ii) a second hybridization between Benthochromini and Perissodini.

Patterson's $D$ tests were performed on all possible permutations of four individuals belonging to each of the three test groups and gene flow interpreted from distributions of $D$-statistics (Fig. 3). We found a strong signal for gene flow between Steatocranini and the ancestor of 'modern' tribes in Lake Tanganyika (all tribes except Bathybatini and Boulengerochromini) shown as negative $D$ values ($Z$-score > 3; Benjamini-Hochberg-adjusted $p < 0.05$). This hybridization signal was consistent whether Bathybatini or Boulengerochromini were used as P2 and regardless of the used outgroup taxa, whereas it clearly disappeared when Steatocranini (P3) was replaced by other close non-Tanganyikan lineages (i.e., Heterotilapiini or Hemichromini) (Fig. 3b and Supplementary Table 3). We also found strong support for the second hybridization hypothesis between Benthochromini and Perissodini, shown as consistently negative $D$ values ($Z$-score > 3; adjusted $p < 0.05$; Supplementary Table 3).

Based on the observed patterns of gene flow, we hypothesize that the riverine Steatocranini contributed genetic material to the Lake Tanganyika radiation, which is in line with a hybrid swarm scenario. The hybrid swarm theory was proposed as a mechanism that can instantaneously generate novel genetic combinations from standing variation, thus facilitating subsequent rapid radiation[8]. Hybridization overcomes the necessity for new mutations to arise and thereby promotes rapid trait divergence through the generation of so-called transgressive phenotypes with higher responses to disruptive divergent selection[8]. The distribution of $D$-statistics for the introgression between Steatocranini and 'modern' Lake Tanganyika tribes shows stronger signals of gene flow for the H-lineage (in particular Haplochromini) and weaker for Lamprologini (Supplementary Data 2). This suggests that gene flow between Steatocranini and the H-lineage continued after the initial divergence of Tanganyikan lineages, perhaps in geographically structured protolakes. Interestingly, per-locus $D$-tests identified several loci with stronger signal of introgression, which are associated with vision, body coloration, fin

development, gene regulation, and immunity (according to gene ontology annotations; Supplementary Table 4). Also, two genes linked with key cichlid innovations, the gene *sp7* (implicated in jaw phenotypic plasticity[53]) and the opsin *sws1*, had strong introgression signals ($D = -0.74$ and $D = -0.14$, respectively). Together, these lines of evidence suggest differential introgression of genes at the origin of the adaptive radiation in Lake Tanganyika. Previous studies have demonstrated several instances of interspecific hybridization among twigs rather than deeper branches of Lake Tanganyika cichlids, sometimes suggested by cytonuclear discordance[18,23,40,54] but no evidence existed until now for the hybridization between the oldest major lineages (tribes) at the base of the Lake Tanganyika radiation. Intriguingly, comparable hybridization scenarios with riverine lineages have been proposed to boost cichlid radiations in Lake Malawi, Lake Victoria, and the Cameroonian crater lakes[23,24,55].

In addition to testing the two newly proposed hybridization events, we used our dataset to revisit the hybridization events proposed by Meyer et al.[25] that employed a novel approach based on age distribution of gene trees. Meyer et al. proposed introgression of Cyphotilapiini into (a) the common ancestor of the Lake Tanganyika haplochromines (i.e., the Tropheini) and (b) *Pseudocrenilabrus*. Our analyses found that not only Tropheini, but also *Astatoreochromis* and Lake Malawi and Victoria haplochromines showed signs of introgression, but *Pseudocrenilabrus* did not (Z-scores > 3, adjusted $p < 0.05$; Fig. 3c and Supplementary Table 3). These results suggest that a single event of hybridization most likely occurred in the haplochromine ancestor before the colonization of Lakes Malawi and Victoria (Fig. 3c) thus reinforcing the hypothesis of hybridization fueling the adaptive radiations of 'modern' haplochromines[23,24,55]. The role of hybridization in boosting adaptive radiations has long been acknowledged in plants[2], and it is slowly accumulating in adaptively-radiating animals too, including in Darwin finches[3] and narrow-mouthed frogs[56]. Together with narrow-mouthed frogs, our results contribute one of most ancient cases of hybridization described in animals (13.7–12.7 Ma).

Meyer et al.[25] proposed additional introgression events involving Boulengerochromini (or the common ancestor of Boulengerochromini + Bathybatini) and either (i) the H-lineage or (ii) the ancestor of Perissodini + Cyprichromini. Our results show conflicting gene flow patterns whenever Boulengerochromini or Bathybatini are tested (Supplementary Table 3) and thus suggest a much more complex introgression scenario.

**Molecular evolution of genes associated with key innovations.** In order to understand the role and relative importance of ecology and life history in shaping the Lake Tanganyika species flock, we studied the molecular evolution of 29 genes previously associated with key cichlid innovations: bone and tooth development (jaw development), coloration, and colour vision. The link between traits and genes is based on previous investigations including forward genetic studies on model organisms, gene ontology annotations or differential gene expression patterns in cichlids (Supplementary Table 5). Taking advantage of the phylogenetic framework inferred previously (Fig. 1), we studied shifts in the substitution rates (dN/dS) in relation to gene function, phylogeny (lineage effects), ecology, and life-history traits. General measurements of dN/dS (or selection coefficient, $\omega_{M0}$) showed that the three gene categories evolved under different strengths of purifying selection: strongest for tooth and bone development genes ($\omega_{M0} = 0.014$–$0.321$), weakest in colour vision genes ($\omega_{M0} = 0.223$–$0.469$), and intermediate in coloration genes ($\omega_{M0} = 0.105$–$0.463$). All other anchored loci not considered in the previous three categories had a wide range of coefficients of

selection (mean $\omega_{M0} = 0.166 \pm 0.14$) (Fig. 4a). The different mutational target sizes and pleiotropy levels of the three gene classes can explain this pattern: bone and tooth development genes have high pleiotropy (many are housekeeping) and colour vision is determined by a small set of opsin genes with few known pleiotropic effects, whereas coloration genes are intermediate. In addition, colour vision is determined by a small number of genes, which facilitates the identification of changes in selection compared to other quantitative traits such as jaw development that are controlled by many interacting genes of small effect. Six out of 356 'anchored loci' not considered in the three studied categories had relatively high dN/dS ($\omega_{M0} > 0.4$) and these had functions related to reproduction, craniofacial growth, or to digestive and immune systems (Supplementary Table 6).

Thirteen out of the 29 studied genes showed significant evidence of positive selection (detected using M2a/M1a random site models in PAML[57], $\chi^2$-test with 2 df, after Benjamini-Hochberg correction for multiple testing) (Supplementary Table 7). All genes responsible for colour vision were under positive selection and analysed later in detail. Two out of eleven genes associated with bone and tooth development (*c-fos*, *col6a1*) showed signs of positive selection. The differential regulation of these genes in the lower pharyngeal jaw across developmental stages and in experimental studies of plasticity in response to diet[58,59] suggests that they play a role in determining size and shape of the lower pharyngeal jaw, a key innovation tightly associated with food source[60]. Four out of eleven coloration genes previously associated with melanocyte survival, differentiation, and migration (Supplementary Table 7) showed evidences of positive selection: *csf1ra, dlc, kita,* and *kitla*. The *csf1ra* gene was hypothesized to be under positive selection in haplochromines and involved in eggspot formation[61], but our Clade model C (CmC) analyses found that an alternative scenario of rate changes at the origin of the Lake Tanganyika flock and in the Haplochromini was more likely ($\delta AIC = 28.33$; outgroup $\omega = 0.808$, Lake Tanganyika $\omega = 0.047$, Haplochromini $\omega = 1.089$; Supplementary Table 9). The three-rate-change scenario was also favoured for *kita*, which involved in melanophore development, but non-haplochromine Tanganyikan cichlids displayed the fastest rates (outgroup $\omega = 2.834$, Lake Tanganyika $\omega = 6.492$, Haplochromini $\omega = 3.796$; Supplementary Table 9). The gene *dlc*, a major contributor to the formation of stripe patterns in zebrafish, shows an acceleration of positive selection in the Haplochromini (Fig. 4c, Supplementary Table 9; non-haplochromines $\omega = 0.658$, Haplochromini $\omega = 18.527$). The CmC models for *kitla* were not found to statistically deviate from neutral evolution. Even though the M2a/M1a site test was not significant for *hag* (Supplementary Table 7) this gene was previously hypothesized to be under positive selection in haplochromines[62], a result confirmed by our analyses that reveal a significant rate difference (non-haplochromines $\omega = 0.103$, Haplochromini $\omega = 10.945$; Supplementary Table 9). For all five mentioned colour-associated genes, CmC accounting for lineage effects fitted the data better than those accounting for rate shifts associated with the presence of sex-related colour dimorphism (Supplementary Table 9).

Adaptive radiations, which display vast phenotypic diversity in the absence of pronounced molecular differences, might be seen as a laboratory of "natural mutants" that can help identify functional variants of genes, much like mutagenesis screens in model organisms[6,7]. Screening for variants of potentially large effects revealed premature stop codons in *mitfa* and *kita*, two genes essential for melanophore survival that could contribute to xanthic phenotypes (e.g., *Ctenochromis horei*). Premature stop codons were also prevalent in Bathybatini for *dlc* and *hag*, where some naturally occurring species have disrupted barred

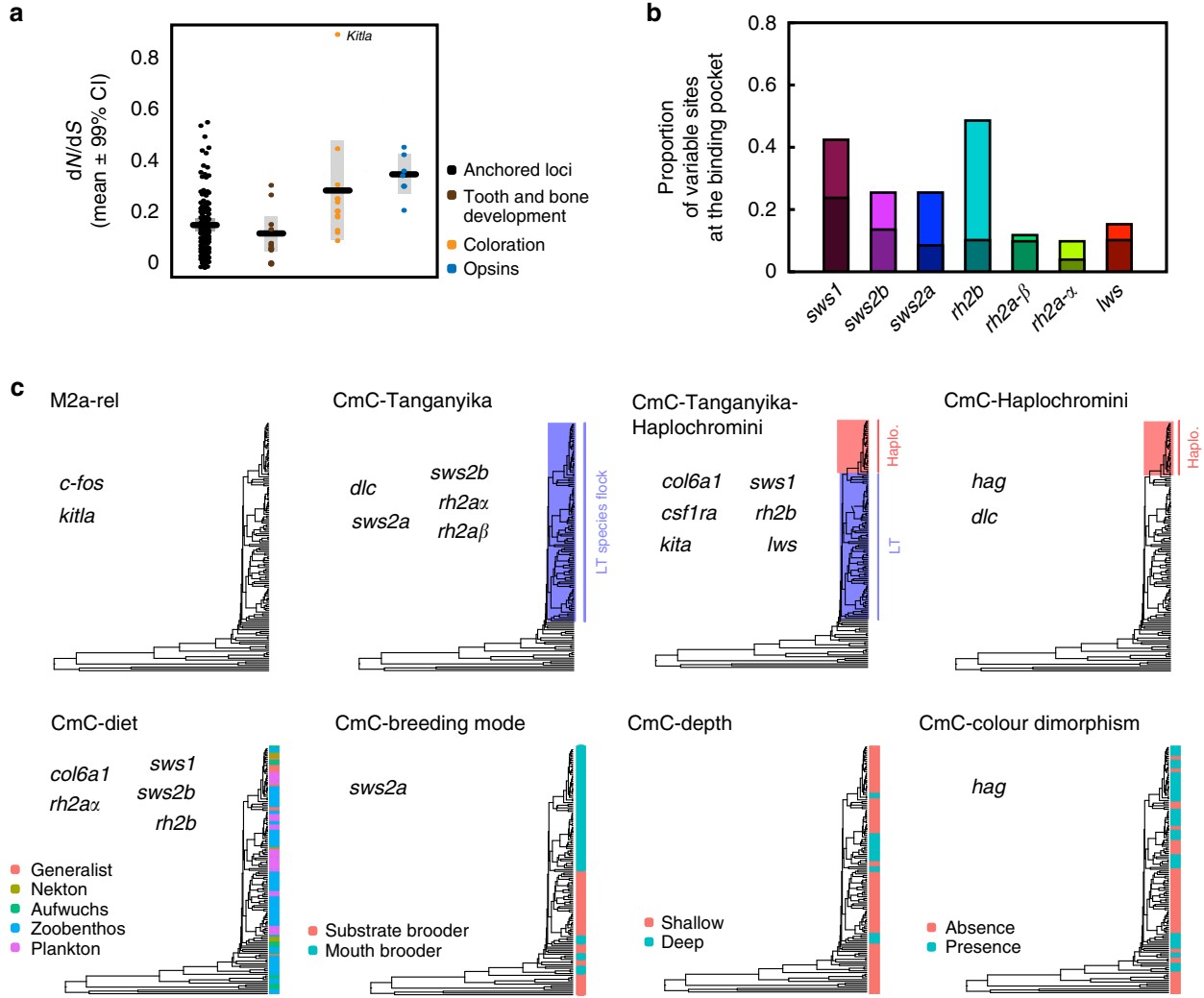

**Fig. 4** Molecular evolution of genes associated with cichlid innovations. **a** Strength of natural selection on genes involved in tooth and bone (jaw) development, coloration and colour vision (cone-opsins) in comparison to the genome-wide pattern represented by the remaining anchored loci. **b** Proportion of variable sites directed into the chromophore-binding pocket of cone-opsin genes. Shaded bars represent the subset of amino acid substitutions that result in changes in polarity. **c** Species partitions used in Clade C model (CmC) tests according to phylogeny, ecology, or life history traits. Gene names are shown under the CmC models with the highest explanatory power (lowest AIC) for both phylogeny- and ecology/life-history-based partitions. LT, Lake Tanganyika; Haplo., Haplochromini

patterns resembling zebrafish knockout mutants of these genes[63].

**Diet is a major force shaping the evolution of opsins.** The molecular evolution of cone-opsins (colour vision) was analysed in further detail. The genotype–phenotype association in cone opsins is straightforward, the phenotypic consequences of amino acid replacements are well understood and extensive work on cichlid opsins[21] provides a solid basis for the interpretation of our results. Sites at chromophore-binding pockets are highly variable in all cone-opsin genes (Fig. 4b) and often under positive selection (Supplementary Table 7). This suggests the occurrence of structural changes in the studied opsins, which can potentially shift spectral sensitivity. However, direct evidence for changes in spectral sensitivity would require microspectrophotometry and functional validation experiments.

Six out of the seven opsin genes showed faster rates of positive selection in the Lake Tanganyika species flock (Fig. 4c and Supplementary Table 10) suggesting that this adaptive radiation

coincided with the diversification of the visual system[64]. For the seventh gene *lws*, a model showing deceleration of positive selection in the Lake Tanganyika flock followed by acceleration in Haplochromini fitted the data best. According to AIC, CmC considering divergent rates associated with ecological traits explained sequence variation better than neutral models in five out of seven opsins (Fig. 4c and Supplementary Table 11). In four out of these five genes, diet had the highest explanatory power among all tested ecological and life history factors (Fig. 4c). For example, planktivorous cichlids display faster rates of evolution for the UV-sensitive opsin *sws1*, in agreement with the observed higher expression of this gene when compared to other cichlids[65]. This finding agrees with the fact that increased UV sensitivity would confer higher efficiency in foraging on plankton, which is translucent except under UV light. In contrast, cichlids grazing on Aufwuchs (algae adhering to open surfaces and other small organisms and organic matter attached to them) show the fastest rates in the green-sensitive opsin *rh2a-α*. CmC allowing different rates in shallow-dwelling versus deep-dwelling cichlids fit data best

for *lws* (absorption $\lambda_{max}$ ~565 nm) and second best for *sws1* (absorption $\lambda_{max}$ ~375 nm), in both cases shallow-dwelling cichlids having faster rates of positive selection. In contrast, deep-water dwellers had higher rates for *sws2b* (absorption $\lambda_{max}$ ~425 nm). This makes sense given that UV and red light are diminished from the light spectrum in deep water[66]. This evidence further agrees with the pseudogenization of *sws1* in the deep-water tribe Bathybatini (all individuals had premature stop codons). Despite notable differences in AIC with other models, CmC partitioning for depth was not significant for *lws* after Benjamini-Hochberg correction. These results, together with previous studies that found divergent evolution in the dim light opsin *rh1* between deep and shallow water Tanganyika cichlids[67], support depth as an important driver in the evolution of colour-sensitive opsins. CmC models accounting for differences between species with and without sex-related colour dimorphism and between substrate- versus mouth-breeding cichlids fit data significantly better than the null model (neutral evolution) only for short wavelength sensitive opsin genes (absorption $\lambda_{max} < 470$ nm). In all cases, sexually mono-morphic and substrate-breeding species had faster rates than sexually dimorphic or mouth-brooding cichlids (Supplementary Table 11). The molecular evolutionary analyses of the 29 selected innovation-associated loci revealed interesting patterns connected to cichlid adaptive radiations, ecology and life history, but the generality of such results will need to be confirmed, refuted, or adjusted using larger genome-wide approaches such as genome re-sequencing.

**Conclusions.** Based on a large anchored phylogenomic data set we were able to produce a robust time-calibrated phylogenetic hypothesis for the cichlid family, with a focus on the Lake Tanganyika radiation, the evolutionary reservoir for all major East African cichlid radiations. Our tree is the most comprehensive so far in terms of phylogenetic diversity, taxa, and number of genes. The divergence times estimated here are in accordance with two major geological events in cichlid history: the radiation of the Lake Tanganyika species flock coincides with the lake formation, and the deepest splits in the cichlid tree agree with Gondwanan vicariance. We find novel evidence for hybridization at the onset of the Tanganyika species flock and suggest this event could have boosted the subsequent adaptive radiation. In support of this hypothesis, we find a strong signal of introgression in genes responsible for colour vision and jaw development, two cichlid key innovations. Using this new robust phylogenetic framework, we examined the molecular evolution of 29 genes associated with key cichlid innovations in order to weigh the relative contribution of lineage effects, ecology, and life history traits. Several of these innovation-associated genes display an acceleration of positive selection that coincides with the origin of the Lake Tanganyika flock, and among the studied ecological traits, diet explained a large portion of the variability. Taken together, our findings contribute to a better understanding of the patterns and processes shaping the adaptive cichlid radiation in Lake Tanganyika and the East African Great Lakes.

## Methods

**Anchored loci sequencing and dataset assembly.** We included cichlid representatives of all major lineages and geographic regions (India 2 sp., Madagascar 4 sp., Neotropics 6 sp., East African Great Lakes 114 sp. and African riverine species 23 sp.). Taxon sampling emphasized the Lake Tanganyika radiation by including all tribes except Trematocarini (no tissue of sufficient quality was available). Detailed information of taxon sampling is available in Supplementary Table 1. Tissue samples were acquired from aquarium trade or wild populations. For the latter, sampling and export was carried out within the framework of a Memorandum of Understanding between the Department of Fisheries, Ministry of Agriculture and Livestock, Republic of Zambia, and the Universities of Zambia at Lusaka, Graz, Bern and Basel (2008–2013). Additional sampling and export was granted by the Ministry of Agriculture of Zambia (permit S1248/08 to C. Sturm-bauer) and the Ministry of Environment and Natural Resources of Nicaragua (permit DGPN/DB/DAP/IC-0003-2012 to A. Meyer). Tissue sampling and export complied with local regulations and applicable ethical standards.

Phylogenomic data were generated at the Center for Anchored Phylogenetics (www.anchoredphylogeny.com) using the anchored hybrid enrichment methodology[68]. Briefly, DNA was extracted from fin clips using the ZR-96 Genomic DNA Tissue MiniPrep (Zymo, CA, USA) and DNA extracts sheared to 150–300 bp using a Covaris E220 Focused-ultrasonicator. Indexed libraries were prepared on a Beckman–Coulter Biomek FXp liquid-handling robot following Meyer et al.[69] with an additional size-selection step after blunt-end repair using SPRIselect beads (Beckman–Coulter; 0.93 ratio beads:sample volume). Indexed samples were pooled at equal quantities (12–16 samples per pool) and enrichments were performed on each multi-sample pool using an Agilent Custom SureSelect kit (Agilent Technologies, California USA). After enrichment, the pools were grouped in equal quantities for sequencing into four paired-end 150 bp Illumina HiSeq2000 lanes. Sequencing was performed in the Translational Science Laboratory, College of Medicine, Florida State University. Paired overlapping reads from fragments <280 bp were merged, sequencing errors corrected, and adapter sequences trimmed off. Reads were assembled using a quasi-de novo approach in an in-house pipeline[70]. Orthologous sets were identified from the assembled consensus sequences using sequence similarity and clustering in a neighbour-joining-like process[70].

Targeted loci correspond to the anchored loci for teleosts[71], designed from individual exons of 260 gene families selected for their good phylogenetic performance and low-copy number. In addition, we supplemented the bait set with 29 genes (176 exons) with functions related to colour vision, coloration, and bone and tooth formation (Supplementary Table 5). Probes for the 29 candidate genes were designed from exon alignments from three African cichlids (*Oreochromis niloticus*, *Metriaclima zebra*, and *Pundamilia nyererei*) using the annotations from the *O. niloticus* genome (Orenil1.0).

In addition to sampled species, homologous sequences from nine available cichlid genomes were added to the alignments of anchored loci: *O. niloticus*[72], *Astatotilapia burtoni*[72], *M. zebra*[72], *Neolamprologus brichardi*[72], *P. nyererei*[72], *Amphilophus citrinellus* (A. Meyer, University of Konstanz), *Andinoacara coeruleopunctatus* (M. Malinsky, Wellcome Trust Sanger Institute), *Petrochromis trewavasae* and *Tropheus moorii* (C. Sturmbauer, University of Graz). Orthologs were identified by reciprocal BLAST searches and the hit coordinates were extended by 500 bp upstream and downstream to capture less conserved regions flanking exons. Individual loci were aligned with MAFFT v.7245[73] and positions with >80% gaps were removed. Eight out of the 541 initially assembled loci were excluded because of suspected paralogy: one locus (KCNJ13) produced BLAST hits in two regions of the *O. niloticus* genome and has been identified as duplicated[72] and the remaining seven loci had a high proportion (>3%) of ambiguous base calls suggesting that consensus sequences might have been assembled from reads originated from several paralogs. Pairwise genetic distances and manual visualization of alignments were used to ensure correct orthology and remove any non-homologous sequence stretch in the added genomes.

**Phylogenomic analyses and estimation of divergence times.** A concatenated maximum likelihood tree was constructed with RAxML v.7.3.1[74] using independent GTR + Γ models per gene and branch support assessed by 500 pseudo-replicates of non-parametric bootstrapping. A coalescent tree was estimated with ASTRAL II[75] with default parameters using maximum likelihood locus trees estimated with RAxML. Branch support for the coalescent tree was assessed by local posterior probabilities and 100 replicates of multi-locus bootstrapping with site resampling. We assessed the degree of conflict at internal branches using internode certainty as implemented in RAxML v.8.1.20 after applying a prob-abilistic correction for partial gene trees[38]. A gene jackknifing analysis[76] was performed to estimate the proportion of final-tree bipartitions recovered by coalescence vs. concatenation as a function of dataset size. 100 jackknife replicates were generated each for sets of 10, 100, 200, 300, 400 and 500 randomly selected loci and separately analysed with ASTRAL and RAxML.

Molecular dating was performed with RelTime[77] as implemented in MEGA-CC v.7.0.20[78]. We used the GTR + Γ with five gamma categories (the prevalent model for single loci identified with jModeltest[79]), assumed the "local clocks" model, set the branch swap filter to "None" and fixed the topology to that of maximum likelihood tree (Supplementary Fig. 1). To obtain robust divergence times and understand the effect of different fossil calibrations, we performed independent analyses using ten calibration schemes: The first scheme (C01) assumed vicariant splits associated with Gondwanan fragmentation. Schemes C02–C03 use five and four fossil calibrations, respectively, and are thus independent of vicariance assumptions. Schemes C04–C05 extend respectively C02–C03 by applying maximum constraints for the nodes affected by vicariance in C01. Schemes C06–C10 correspond exactly to C01–C05 except that they all included †*Tugenchromis pickfordi*[37] (see Supplementary Table 2 for full details on calibration schemes). RelTime uses only minimum and maximum calibration bounds without assuming a specific probability distribution. The commonly used software BEAST proved computationally intractable for this large dataset, which

failed to converge despite long running times on powerful multi-CPU and multi-GPU based servers.

**Hybridization tests**. The signal for hybridization was explored using galled networks, constructed as follows: SplitsTree4[80] was used to generate a filtered supernetwork from 533 unrooted maximum likelihood locus trees using the Z- closure method[81], which accounts for incomplete taxon sampling across loci. After filtering splits supported by <50 locus trees, we estimate galled networks with the Z-closure method in Dendroscope3[82] using various thresholds (10, 15, 20, 25, 30, 40, 50) that correspond to the percentage of locus trees that support the network edges. We used the galled network with reticulated edges supported by ≥20% locus trees (Supplementary Fig. 4) to formulate hypotheses of hybridization because it represents a good trade-off between locus tree support and total number of hybridizations. Following this strategy, two inter-tribal hybridizations were hypothesized and further tested using the Patterson's D-statistic[52] as implemented in the evobiR R package (https://github.com/coleoguy/evobir) following Durand et al.[83]. In addition, we revisited hybridization hypotheses proposed by Meyer et al.[25]. For each hypothesis, we selected three relevant lineages and one outgroup conforming a four-taxon asymmetric tree required by D tests, following the maximum likelihood topology (Supplementary Fig. 1) and ensuring all hypothesized gene flow events could be tested. Patterson's D tests were performed on all possible permutations of individuals belonging to each of the four test groups, generating distributions of D-statistics (summarized in violin plots). To measure the significance of these tests, Z-scores and p-values were calculated from 100 bootstrap replicates of the tests. P-values were adjusted for multiple testing using the Benjamini-Hochberg method. For the main hybridization hypothesis (Fig. 3b) per-gene D tests were carried out by pooling together all relevant taxa into four test groups (LTm, Bou + Bat, Ste, Til; *sensu* Fig. 3b); loci with too few (<5) ABBA or BABA sites were filtered out, and the loci with strong hybridization signal were annotated using gene ontologies (derived from *O. niloticus*[64]) and summarized with REVIGO[84]. To avoid possible biases in base calling due to heterozygosity, all D tests were performed on bioinformatically-phased haplotypes[85], randomly choosing one allele per individual and locus. To rule out biases due to outgroup choice in D tests, these were repeated with different outgroups. The summary of all D tests is available in Supplementary Table 3 and the detailed information of all permutations in Supplementary Data 2.

**Molecular evolutionary analyses of innovation-linked loci**. Coding sequences from 176 individual loci alignments were extracted and concatenated into 29 gene alignments using the *O. niloticus* genome as reference. Any sequence containing premature stop codons within the ORF was excluded. Gene alignments for *rh2a-α* and *rh2a-ß* were derived from bioinformatically-phased alleles[85] because the default assembly pipeline could not disentangle these close paralogs. We analysed dN/dS using random site models in PAML v.4.8[57]. Likelihood ratio tests (LRT) were used to calculate significance of random site models testing for dN/dS variation across sites (M3/M0) and positive selection (M2a/M1a) and in the latter case, sites under positive selection were identified using the Bayes' Empirical Bayes method in PAML and compared results to those obtained by SLAC, MEME, FUBAR, and FEL methods in HYPHY[86]. Using Clade model C (CmC) in PAML, we tested lineage effects and ecological and life history factors as drivers of molecular evolution, similarly to previous work in Neotropical cichlids' *rh1* gene[64]. CmC allows rates of positively selected sites to vary among a priori defined groups of species (background vs. foreground branches). Significance of CmC was tested against a null model of neutral evolution (M2a_rel[87]) by LRT and the relative fit of the different CmC models assuming different taxa partitions were compared by AIC. The following lineage-specific partitions were used: (i) Tanganyika species flocks vs. all other cichlids, (ii) Haplochromini vs. all other cichlids, and (iii) Tanganyika-Haplochromini, similar to (i) but assuming independent rates for Haplochromini (Fig. 4c). To test variation associated with life history or ecological factors in opsins, the analyses were centered on the Lake Tanganyika flock because they showed markedly faster rates compared with the outgroup. Taxa partitions for ecology and life history traits were defined upon ancestral character state reconstructions in order to define partitions more objectively. First, individual species were classified according to (i) diet (zoobenthos, nekton, plankton, Aufwuchs, or generalist), (ii) habitat (deep-water or shallow-water), (iii) breeding mode (substrate- or mouth-brooders), and (iv) presence or absence of sexual colour dimorphism (Fig. 4c). Ancestral states were inferred for each character (Supplementary Figs 5–8) using the re-rooting method as implemented in the R package phytools[88]. Coding of traits in terminal taxa followed a number of resources, including literature, digital databases (www.fishbase.org), and field observations (S. Koblmüller). To minimize the number of transitions and avoid stochastic errors of single gene trees, ancestral reconstructions and dN/dS calculations assumed the maximum likelihood topology (Supplementary Fig. 1) and only changes with a marginal posterior probability >0.85 were considered. Analyses based on gene trees did not qualitatively change the obtained results. All tests were adjusted for multiple testing using the Benjamini-Hochberg procedure.

**Data availability**. The final datasets are available in Figshare (https://doi.org/10.6084/m9.figshare.6182519).

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

## Acknowledgements

We thank Ulrich Schliewen for providing key tissue samples and Milan Malinsky for
allowing access to the *Andinoacara* genome. We are grateful to Christine Börger, Lénia
C. Beck, and Daniel Monné for laboratory assistance. We thank Michelle Kortyna, Sean
Holland, Alyssa Hassinger, and Kirby Birch at the FSU Center for Anchored Phyloge-
nomics for assistance with data generation and analysis. We appreciate the support of
Nikos Chararas (Graz University of Technology) and Ursula Winkler (Karl-Franzens-
University Graz) in the use the High-Performance-Computing infrastructure. We are
grateful to Ricardo J. Pereira and Heath Blackmon for discussions about *D* tests and to
Julia Jones for proofreading. This work was supported by the Deutsche For-
schungsgemeinschaft (DFG) (AM), the European Research Council (ERC Advanced
Grant "GenAdapt" No. 273900 to A.M.), the Austrian Science Fund (Grant P22737 to C.
S.), the Austrian Ministry of Science, Research and Economics (OMICS Center HSRSM
initiative to G.G.T.) and the University of Graz (Incentive grant to S.K.). I.I. was sup-
ported by postdoctoral fellowships from the Alexander von Humboldt Foundation
(1150725) and the European Molecular Biology Organization (EMBO ALTF 440-2013).
P.S. was funded by a PhD scholarship from the Austrian Centre of Limnology. F.H. was
funded by the Brazilian National Council for Scientific and Technological Development
(PDJ 406798/2015-0).

## Author contributions

Conceived study: C.S., A.M. Fieldwork: S.K., A.M., C.S. Sampling design: S.K., A.M., C.S.,
F.H., P.F., I.I. Laboratory work: P.F. Anchored loci: E.M.L., A.R.L. Phylogenomics and
hybridization analyses and manuscript draft: I.I., P.S., S.K. Divergence time estimation:
G.G.T., C.F., S.K. Functional loci analyses: J.T.D., F.H., I.I., S.K., P.F. All authors con-
tributed to and approved the final manuscript.

## Additional information

018-05479-9.

**Competing interests:** The authors declare no competing interests.

**Reprints and permission** information is available online at http://npg.nature.com/
reprintsandpermissions/

**Publisher's note:** Springer Nature remains neutral with regard to jurisdictional claims in
published maps and institutional affiliations.

