## [Peer Review File · Nature Communications]

Reviewers' comments:

Reviewer #1 (Remarks to the Author):

This study by Irisarri et al provides additional data and analyses towards exploring the cichlid radiation of the Rift lakes. The study sequenced several new populations using a targeted subset of genes previously identified as potentially functionally connected to adaptive phenotypes. This targeted gene set is the strength of the study, and allows the authors to have a large amount of data to analyze rates of molecular evolution within these pre-identified categories of genes that had previously been identified as adaptive. However, the use of an explicitly biased set of genes (specifically ones in part identified by their accelerated molecular rate) raised for me many questions about their suitability as 'unbiased' estimators for molecular rates, D-statistics, and even gene tree distributions, for which having loci are sampled in an reasonably unbiased manner from a variety of molecular rate categories would be desirable. I also have several specific questions in particular about the interpretation of the D-statistics and PAML analyses. Finally, there were several typos found (see a few in the minor points below). The findings here are also largely supportive or critical of previously established hypotheses about cichlids and I found no new methods specific to this study, and therefore I found the overall novelty moderate. If these questions can be resolved, I think this study could make an substantive addition to the literature in continuing to phylogenomically deconstruct this complex group.

Major Questions:

1) The choice of dataset

The authors specifically mention using a set of "adaptive" versus "neutral" loci. The criteria for annotating these gene sets appears to be based on previous literature. Were the criteria for these categories originally based on GO terms, or were these curated from specifically genes that were identified as being adaptive based on high rates of molecular evolution. I first wondered whether the authors think this could have explained some of the differences in rates of evolution observed between this and other datasets, since putatively "adaptive" loci were used? Could this have affected the D-statistic measures or PAML results? I would recommend the authors explicitly address this use of a specifically curated and designed dataset (rather than an naive whole-genome or randomized RAD dataset) and how it might affect the results of the rate estimation, dN/ds and D-statistics more specifically.

2) Questions about the D-statistic

I have the most questions about your methods and conclusions for the D-statistics shown in Figure 3:

- The trees presented in Figs 1, 2, S1 and S2 all show different arrangements of the four taxa used in Figure 3b. While this is indeed evidence for possible ILS/introgression, an explicit argument was not made as to why this particular configuration was chosen for the D-statistics.
- Were the D-statistics as extreme in value or significance when other tree topologies were chosen?

- Did "BBAA" patterns substantially outnumber "ABBA" and "BABA" (as would be assumed under this model)? What is the average raw number of BBAA/ABBA/BABA sites? Without understanding the absolute values of these, the ratios are hard to interpret. (even with the summary data in Supp Table 13)

- For Figure 3b itself, why are the two groups lumped together on the left? This seems redundant since it is merely the visual addition of the two violins shown on the middle and on the right (correct?).

- On II. 344-345, a popD value of 0.017 is reported and the D-statistics violin is roughly split between positive and negative values. While statistically this may be technically significant under the Z-score, this means a claim of introgression is being based on a 1.7% skew in ABBA vs BABA, which is quite small in actual magnitude. Also, to me a distribution of large-magnitude values of D that are both positive and negative does not imply that there is no introgression, but rather that different populations are showing different evidence for introgression. A finding of generally no introgression would manifest as a unimodal distribution concentrated at $D=0$, not a bimodal $D\pm$ distribution.

- Also in II.391-393, there are other D-values reported that are between 0.01-0.06. Again, this means an excess in ABBA/BABA skew of $> 6\%$. While the Z-score test is significant (but see below), this is still quite a small effect size, or could be evidence of multiple introgressions (since introgressions of both P3-P1 and P3-P2 simultaneously will dilute the magnitude of D).

- Given the shape of the distribution for Bathybatini, this might indicate that certain individuals are introgressing with different groups? Was this examined?

- Was introgression localized in certain loci?

- Was bootstrapping done over individuals, loci, or both? The original usage was jackknifing over genomic windows, why was bootstrapping used here instead? Furthermore, while Z-scores have been common practice in D-statistic, I still do not understand fundamentally in any study where it is used what this tells you about the D-statistic measure. Under what conditions would a D-statistic Z-test fail to be significant? What is the alternative? Why does permuting the data inform the presence/absence of introgression overall? Again, I understand that Z-scores are the standard from Patterson's original use in Neanderthals, but I am still (after many papers on the subject) unclear as to why resampling and Z-scores are informative. (Also this falls prey in some ways to the effects of large datasets in bootstrapping, where small differences are magnified by the scale of data).

- Given your species configurations, you might consider the newer DFOIL test of introgression (Pease & Hahn 2015 Syst Biol) that uses pairs of taxa and purports to show directionality of introgression.

3) Questions about PAML Analyses

- On I. 410 the authors call the phylogenetic framework "robust," which implies that the tree is generally suitable for testing dN/dS for all genes. However, Figure S2 clearly shows substantial gene tree discordance, so I am unclear why the framework is "robust" for gene-by-gene testing.

- On I. 416 what does "markedly different" mean specifically? Are the distributions

statistically different? They seem to overlap with each other and the background distribution.

- While the targeted sequencing of putatively "adaptive" loci was an effective way to increase sampling depth, I am unsure how to interpret these results or draw general conclusions about the entire genome. It seems more than plausible that other loci might have been under even stronger selection that were simply not sequenced, so I would recommend recontextualizing the results in the last section about diet to note that this result is only reported within the context of the pre-selected, targeted genes sequenced here and do not represent a naive genome-wide or random RAD-set of loci.

- Also recommend being (even more) specific about what is meant by "neutral"? As in "not under recent selection post-radiation", etc.

Minor Points:

- I realize it is still conventional (unfortunately, in my opinion) to use bootstrap/PP the branch score annotations for the primary figures and relegate IC scores to the supplement. However, the problems (and lack of information) with bootstrap/PP for large-scale data have been demonstrable in nearly every recent phylogenomic study of the last 5-10 years. I would encourage the authors to consider at least including the IC scores alongside the bootstrap/PP in Figure 1 or even swapping the scores used in figures 1 and supp fig 2. To me supp. figure 2 contains far more information that is useful than the presented Fig 1 tree that is almost completely perfect scores (when in fact many of the sub-familial branches have quite low IC scores). This argument is made verbally, but I think it would be impactful to show this in graphical form in the main paper. Depending on one's view of IC scores or the limitations of bootstrap in large datasets, this may be viewed as a rational or radical suggestion. Therefore, I am putting it in the "minor points" and I leave it to the authors' and editor's judgment.

I. 114 "modeling"

II. 169, 332 and 352: "Patterson" misspelled and "D" should be italicized or math-type

I. 204: "basal" here is not necessary (and misused), recommend rephrasing to "B+B share a common ancestor distinct from other African..."

I. 210: "also" is redundant

I.404: "targeted"

I. 407: What does "neutral" mean here?

I. 417: "toorh"

- [CH Martin et al 2015 Evolution] seems like a notably absent citation as one of the most comprehensive refutations of Cichlid sympatric speciation.

Reviewer #2 (Remarks to the Author):

The authors used a targeted capture and sequencing of about 1 Mb of aligned sequence from 149 species of Tanganyika and related worldwide cichlids with only 9% missing data. They use this data for 1. constructing a well-resolved species phylogeny, 2. time the phylogenetic events, 3. infer basal hybridisation events by D statistics, 4. survey natural selection on ecologically relevant genes in the targeted regions. They conclude 1. that phylogenetic timings fit biogeography very well, 2. that hybridisation occurred early and likely provided important genetic variability for diversification, and 3. that positive selection has been prevalent on opsin genes. I agree with 1. and 3. but am worried about 2., see below

In general, it is a well-designed study, competently carried out and well written and understandable even for a non-cichlid expert as myself. I think that the results are important for our understanding of the cichlid diversification globally and in Tanganyika in particular as a perhaps "easier" situation to solve than the later Malawi/Victoria radiations. The illustrations are generally nice, though a bit cramped. It would help an specialist if geographical location was marked for each species in Figure 2 and Figure 4 has to be made larger to be readable.

I have one important concern: As I understand it sequence reads are trimmed and aligned to a lake Malawi cichlid reference (Maylandia Zebra) and the consensus sequence is then called for each species. This might easily cause reference biases and such biases will depend both on the polymorphism level within each individual tested and its phylogenetic distance to the reference (which unfortunately is not the same for all individuals since the reference is an ingroup). When D statistics are used in e.g. primate research, it is common to use mapping to a species that is an outgroup to the set of species tested. I fear that species closer to the reference have a larger chance of being artificially closer to the reference by this approach because it is likely that the reference allele will be called in the consensus whenever the investigated individual is heterozygous (because such reads will map better). D statistics are VERY sensitive to even the smallest biases so my fear is that some of the hybridisations discussed might be artefacts. Perhaps it is not the most likely, but given the importance of such findings, we need to be surer. I suggest that the authors do at least the following two sets of analyses

1. Use the same reference as before but calling all the polymorphisms within individuals instead of a consensus. Then D statistics can be rerun but now using a random haplotype from the diploid genotype. I think results will change but do not know how much. It would also be great to have measures of heterozygosity for each individual, and this could be used to see whether shared polymorphisms are prevalent and how far back. This could also feed positively into the dn/ds analysis that would be sensitive to a too high level of shared polymorphisms
2. Use a different reference from a different part of the phylogeny and repeat the complete analysis and show that the D statistics are robust to this. If available, an outgroup reference would be preferable, but if not, it should be a reference which is in a different part of the phylogeny with respect to the hybridisations inferred

Reviewer #3 (Remarks to the Author):

The authors aim to understand the evolutionary history relating to the deep structure of the Lake Tanganyika radiation of cichlids, given the poorly understood interrelationships among tribes. The goal is to understand, in particular, the origin and evolution of key innovations and the role of hybridization in shaping the adaptive radiations of the cichlids. To this end, they use an anchored hybrid enrichment approach on the Lake Tanganyika species flock, generating a robust time-calibrated tree of cichlids for use as framework to study patterns of molecular evolution.

The premise for the paper is intriguing and important. The authors lay out the arguments clearly, and explain the very obvious need for the work and the rationale for their approach. I thoroughly enjoyed reading the manuscript and see it as an important and very valuable contribution.

For improving the manuscript, there were two general concerns that came up:

1. Context. The study focuses entirely on cichlids. Starting from the first line of the introduction, almost every sentence in the text is related to cichlids and cichlids alone. While cichlids might be charismatic, the importance of such studies must still be placed within a bigger picture context. In what other lineages has adaptive diversification occurred in a similar way? In what others has adaptive admixture played a role? Why is this of general importance beyond cichlids?

2. Hypothetical framework: It would help to have some hypotheses that could be tested. I think that this would make the arguments stronger. At the moment, the study is set up to "demonstrate the power of the anchored hybrid enrichment strategy", which makes it seem like a Methods paper. It could be set up in the context of "We set out to address xx hypotheses in the context of the adaptive radiation of Lake Tanganyika cichlids: (1) timing will coincide with xxxx; (2) hybridization between xxxx might be expected based on findings in Lake Victoria cichlids; alternatively ... (3) selection tends to act most strongly on" etc. Such a framework would be straightforward, and the authors already have hypotheses for some elements, though buried in the Results (eg, lines 264-266 outlines alternative scenarios for timing; lines 377-390 start to talk about possibilities for hybridization scenarios based on previous studies; and lines 471-474 give the beginnings of hypotheses for the role of selection on opsin genes).

Minor comments:

Line 151 "approaches now permit to generate large" Fix ("approaches now permit the generation of large"?)

Line 158 "has demonstrated to enhance " Fix

Line 404 "Anchored sequencing of targeted genes" Fix

Lines 531-533. This last sentence makes the paper come off as methods focused. The results are really interesting, and it would make the work stronger to focus on what the methods have shown rather than the methods themselves.

Figure 2. is not at all clear. If the vertical blue shadows represent splits of Gondwanan-derived landmasses and the formation of Lake Tanganyika, which one represents what? The

superimposed maps don't help much. It would help to mark on the tree the colonization of the lakes.

Reviewer #4 (Remarks to the Author):

The manuscript presents a study of the inter-tribal hybridizations in cichlids using target sequencing to obtain a large number of loci. The selective pressure on some of these loci is also investigated to characterize the adaptive evolution in the cichlids.

The manuscript is well written, but the text could be made more concise to help the reader see the main aspects of the results. At the same time, some analyses are lacking details, which leads to potential issues with these analyses.

First, the manuscript is quite lengthy and the authors should attempt to propose a more concise text. It will help the reader to understand the key elements that are of interest for a wider audience without being lost in the details. For instance, the introduction is long and the part on the timescale of cichlids evolution could be merged with the key innovation part and shortened. Similarly, the first part of the results and discussion on the anchored phylogenomics should be shortened as it is not the most interesting for non specialists in cichlids.

Second, the study hypothesizes that hybridization events and the presence of ILS are affecting the evolutionary history of cichlids. It is thus surprising that a standard concatenation analysis returns the same well supported phylogenetic tree as a coalescent tree (lines 181). This would suggest that the evolutionary history was not affected that much by these events. I would like the authors to comment and discuss this aspect.

Third, the divergence time analyses are really far from being optimal. I can understand that with the large amount of data presented here, some complex analyses using BEAST might not be very practical, but the lack of convergence in the analyses are probably more due to the large heterogeneity in the rate of evolution present in the data than anything else. Indeed, it is evident from figure 2 that including the basal lineages in the analyses drastically increases the range of evolutionary rates, which, combined with the short rates in the cichlids, will then be difficult to model using a single distribution. Using state-of-the-art method is essential to have reliable estimates and to make sure that the different dates obtained compared to Matschiner et al are due to the larger data rather than methodological issues. It would be thus important to reduce the data to focus on the main cichlid tribes and thus reduce the heterogeneity in rates that is causing the lack of convergence. You should also clarify how the uncertainty in the calibration points were incorporated in the analyses (248-249). This is critical especially as the dates obtained differ from some in the literature.

Fourth, the criteria used to select the set of genes for the target sequencing are unclear. The authors should provide a clear description of the procedure used and which gene

regions were targeted and why. You further say once that you have 260 + 29 gene regions (line 555-560) while you talk about "533 aligned anchored loci" (line 177). Could you clarify how these two numbers were obtained?

Fifth, I do not understand the arguments given in lines 260-266. Why do you argue that an initial divergence in the river system would lead to tribes having broader niches? This is entirely dependent on the type of ecological niches that you will have in the different river system. You could have very similar niches in these river systems, which would lead to narrow niches for the different tribes. In any case, I am not sure that you have the data to show this and this argument does not really hold.

Sixth, the hybridization and ILS events are investigated using Patterson's D, but it is unclear how the authors set up the topological constraints to perform these tests. It would be good to have more details about the expectations tested. On a similar note, the groups tested include different species, and not population level data as in the standard test. How did you deal with the potential substitutions occurring within the groups formed? Did you remove these sites before applying the D test?

Finally, I am a bit puzzled by the selection test done and more specifically the significance level used. You tested 29 genes and each gene has a certain probability to be significant. You are thus clearly in a multiple testing problem and the threshold for the p-value should be adjusted (see for example Studer and Robinson-Rechavi, 2008). Further, the M2a/M1a test assumes that the sites are under selection across the whole phylogenetic tree, which in your case means that it is also under selection in the basal lineages of the phylogenetic tree. Did you include these basal lineages in the analyses and, if yes, how can you interpret these results as adaptation interesting for the Lake Tanganika cichlids? The clade model will give you this information, but not the site model. You also argue that ecological traits explain sequence variation better than null models (line 491). How did you do these analyses exactly? Did you run the clade test on several partition of the data and took the one that best explained the evolution of the gene? How exactly and did you correct for multiple testing? This should be clarified and the details of the analyses given as it is a key result of the manuscript.

Minor comments

- line 78: change to "pertains to the deep"

- line 102: it is well known that even with full genomic data, we will not be able to estimate the true species tree without some errors. Using a cloud of trees will then always be necessary.

Response to Reviewers' comments

Reviewer #1:

This study by Irisarri et al provides additional data and analyses towards exploring the cichlid radiation of the Rift lakes. The study sequenced several new populations using a targeted subset of genes previously identified as potentially functionally connected to adaptive phenotypes. This targeted gene set is the strength of the study, and allows the authors to have a large amount of data to analyze rates of molecular evolution within these pre-identified categories of genes that had previously been identified as adaptive. However, the use of an explicitly biased set of genes (specifically ones in part identified by their accelerated molecular rate) raised for me many questions about their suitability as 'unbiased' estimators for molecular rates, D-statistics, and even gene tree distributions, for which having loci are sampled in a reasonably unbiased manner from a variety of molecular rate categories would be desirable. I also have several specific questions in particular about the interpretation of the D-statistics and PAML analyses. Finally, there were several typos found (see a few in the minor points below). The findings here are also largely supportive or critical of previously established hypotheses about cichlids and I found no new methods specific to this study, and therefore I found the overall novelty moderate. If these questions can be resolved, I think this study could make an substantive addition to the literature in continuing to phylogenomically deconstruct this complex group.

We sincerely thank the reviewer for all the thoughtful comments and we hope to have successfully addressed all her/his concerns in the new revised manuscript and our responses below.

Major Questions:

1) The choice of dataset

The authors specifically mention using a set of "adaptive" versus "neutral" loci. The criteria for annotating these gene sets appears to be based on previous literature. Were the criteria for these categories originally based on GO terms, or were these curated from specifically genes that were identified as being adaptive based on high rates of molecular evolution.

The initial description of loci as "adaptive" versus "neutral" was misleading and thus has been removed from the text. Loci referred to as "adaptive" have been associated with key cichlid adaptive traits in previous studies, but were not generally shown to be under purifying selection (in 27/29 genes). We now refer to these genes as "associated with key cichlid adaptive traits". The association of loci with adaptive traits is based on previous literature including forward genetic studies in model organisms, gene ontology annotations and gene expression patterns, which has been now clarified in the main text (references to previous literature were available in Supplementary Table 38).

I first wondered whether the authors think this could have explained some of the differences in rates of evolution observed between this and other datasets, since putatively "adaptive" loci were used? Could this have affected the D-statistic measures or PAML results? I would recommend the authors explicitly address this use of a specifically curated and designed dataset (rather than an naive whole-genome or randomized RAD dataset) and how it might affect the results of the rate estimation, dN/ds and D-statistics more specifically.

Anchored loci have been primarily selected for their good phylogenetic performance (see also Lemmon et al. *Systematic Biology* 5:727-744, 2012). Anchored loci contain a conserved core flanked by less conserved regions, typically exons and introns, respectively, although not exclusively. Therefore, all loci contain both coding and non-coding DNA and sites with various evolutionary rates. In comparison to RNAseq and RADseq data, anchored loci will contain respectively lower and higher proportions of non-coding regions than these datasets. In any case, the use of coding regions should not have a big impact in D-statistics because (i) in exons, which evolve in general under purifying

selection, fewer sites will be polymorphic and thus not informative for D-statistics, and (ii) in cases of positive selection, one would expect AABB or BBAA patterns, which again do not influence D-statistics. Therefore, the use of anchored loci is not expected to bias D-statistics. Given the new focus of the paper, the discussion of different dataset types (anchored loci, RAD-seq, or RNA-seq) and their performance in D-tests or dN/dS analysis (not possible for RAD-seq) is beyond the scope of the paper and decided not to include this comparison for the sake of brevity.

2) Questions about the D-statistic

I have the most questions about your methods and conclusions for the D-statistics shown in Figure 3:

- The trees presented in Figs 1, 2, S1 and S2 all show different arrangements of the four taxa used in Figure 3b. While this is indeed evidence for possible ILS/introgression, an explicit argument was not made as to why this particular configuration was chosen for the D-statistics.

We present two highly congruent phylogenetic hypotheses based on ASTRAL (Fig 1) and RAxML (Figs. 2, S1 and S2). The taxa configurations for D-tests were always chosen following the RAxML tree and the different groups varied in order to test different possibilities of gene flow. For Fig. 3b in particular, we used the taxa configuration of the RAxML tree because, unlike the ASTRAL topology, it allowed us to test the gene flow between Steatocranini and 'modern' Lake Tanganyika tribes. This is not explicitly mentioned in the text and the caption of Fig. 3.

- Were the D-statistics as extreme in value or significance when other tree topologies were chosen?

In order to assess the robustness of our results, we have performed additional D tests varying the outgroup as well as several of the test groups and found congruent results with comparable D values and significance.

- Did "BBAA" patterns substantially outnumber "ABBA" and "BABA" (as would be assumed under this model)? What is the average raw number of BBAA/ABBA/BABA sites? Without understanding the absolute values of these, the ratios are hard to interpret. (even with the summary data in Supp Table 13)

The absolute numbers of total informative sites and the number of ABBA/BABA sites for all performed tests are available in Supplementary Tables 13-36.

- For Figure 3b itself, why are the two groups lumped together on the left? This seems redundant since it is merely the visual addition of the two violins shown on the middle and on the right (correct?).

The reviewer is right and the new version of Fig. 3 does not contain that plot.

- On ll. 344-345, a popD value of 0.017 is reported and the D-statistics violin is roughly split between positive and negative values. While statistically this may be technically significant under the Z-score, this means a claim of introgression is being based on a 1.7% skew in ABBA vs BABA, which is quite small in actual magnitude. Also, to me a distribution of large-magnitude values of D that are both positive and negative does not imply that there is no introgression, but rather that different populations are showing different evidence for introgression. A finding of generally no introgression would manifest as a unimodal distribution concentrated at D=0, not a bimodal D+/- distribution.

The new D tests based on phased data (see below) do not show such a bimodal distribution of D values. In addition, the "population-level" D tests where all species were lumped together into a single test have been excluded from the manuscript because they were misleading and their results did not capture the real variation present in the data. Instead, the new "individual-level" D analyses based on phased alleles show much clearer patterns. Instead of mean D-statistics that were misleading, gene flow is now interpreted from the distribution of D values summarized in violin plots and significance measures, as stated in the text.

- Also in ll.391-393, there are other D-values reported that are between 0.01-0.06. Again, this means an excess in ABBA/BABA skew of > 6%. While the Z-score test is significant (but see below), this is still quite a small effect size, or could be evidence of multiple introgressions (since introgressions of both P3-P1 and P3-P2 simultaneously will dilute the magnitude of D).

As specified above, no mean D-statistics are reported and moreover, this signal for introgression between P2-P3 vanished with the use of phased alleles.

- Given the shape of the distribution for Bathybatini, this might indicate that certain individuals are introgressing with different groups? Was this examined?

The new analyses show no gene flow involving Bathybatini. Interestingly, the new analyses show different strengths for introgression signal between Steatocranini and the 'modern' Lake Tanganyika tribes, with a clear weaker signal for Lamprologini and stronger signal for Haplochromini (including Tropheini). This is now highlighted in the text.

- Was introgression localized in certain loci?

We thank the reviewer for raising this interesting question. We have analysed this aspect and found several interesting genes with stronger introgression signal, which might be related to vision, colouration, gene regulation, or behaviour according to GO annotations. In addition, two genes associated with key cichlid innovations also showed strong signals of introgression. This has been discussed in the manuscript.

- Was bootstrapping done over individuals, loci, or both? The original usage was jackknifing over genomic windows, why was bootstrapping used here instead?

We performed bootstrapping over loci because it is the appropriate resampling procedure for mostly unlinked SNPs. As pointed out by the reviewer, jackknifing is run over windows across the genome and this resampling is done by removing adjacent linked SNPs. Because anchored loci are unlinked bootstrapping is more appropriate (Durand et al. Mol Biol Evol 28: 2239–2252, 2011). In fact, we recalculated the significance of several tests with jackknifing but obtained highly inflated Z-scores and p-values.

Furthermore, while Z-scores have been common practice in D-statistic, I still do not understand fundamentally in any study where it is used what this tells you about the D-statistic measure. Under what conditions would a D-statistic Z-test fail to be significant? What is the alternative?

The Z-score measures the dispersion of D values. In ABBA/BABA tests, a difference of more than 3 standard deviations ($z\text{-score} > 3$) corresponds to a p-value of 0.0027, which is generally considered as significant (e.g., Reich et al. 2011. Denisova Admixture and the First Modern Human Dispersals into Southeast Asia and Oceania. Am. J. Hum. Genet. 89:516-528). A Z score would fail to be significant if the bias of ABBA or BABA were <3 standard deviations. In addition to the Z score, we also obtained a P-value (which were corrected for multiple testing using the Benjamin Hochberg method) of confidence in the D statistic for each test using bootstrapping. In relation to Z-scores, our tests had high enough numbers of SNPs in all tests (Supplementary Tables 13-36), which is crucial for obtaining reliable Z-scores. Too few SNPs could increase the variance of D, thus artificially inflating estimates of D and Z-scores.

Why does permuting the data inform the presence/absence of introgression overall?

The distribution of D values from "individual-level" D tests with taxa permutations provides a better understanding of gene flow between the test groups, including the dispersion of D-statistics and the presence of any heterogeneous signal of introgression in the data (e.g., stronger signal of introgression

between Steatocranini and Haplochormini than Lamprologini, see above). By contrast, “population-level” D tests lump together all taxa in each of the 4 test groups and provide a single D value that cannot represent the complexity present in the data and can be misleading, and thus have been removed in the revised manuscript. In addition, sites with ambiguous positions are filtered out from the tests and thus taxa permutations can use a higher number of ABBA/BABA sites than a D-test based on lumped taxa where more sites are lost due to ambiguity.

Again, I understand that Z-scores are the standard from Patterson's original use in Neanderthals, but I am still (after many papers on the subject) unclear as to why resampling and Z-scores are informative. (Also this falls prey in some ways to the effects of large datasets in bootstrapping, where small differences are magnified by the scale of data).

We share the reviewer's concerns. However, we believe that our results are robust, particularly after the use of phased data and individual-level D tests with varying ingroup and outgroup taxa. We now provide not only Z-scores but also Benjamini-Hochberg-corrected p values. Moreover, the high number of SNPs in our tests and the low standard deviation of D suggest that our new Z-scores are not inflated.

- Given your species configurations, you might consider the newer DFOIL test of introgression (Pease & Hahn 2015 Syst Biol) that uses pairs of taxa and purports to show directionality of introgression.

We thank the reviewer for this interesting suggestion. DFOIL is a very appealing particularly to test the first hybridization hypothesis (Fig 3b) because it would allow simultaneously testing gene flow between all four lineages of interest (Steatocranini, Bathybatini, Boulengerochromini, 'modern' Tanganyika tribes) plus an outgroup. Compared to Patterson's D, DFOIL requires a symmetric tree – ((P1,P2)(P3,P4)P0)– and that divergence between P1-P2 < divergence P3-P4. The required taxa configuration could be fulfilled assuming the ASTRAL topology, but this configuration would not allow us to test for gene flow between Steatocranini and the remaining Lake Tanganyika tribes (they are sister taxa), which is the main hypothesis we want to test. In addition, it is unclear whether the assumption of divergence times would hold.

3) Questions about PAML Analyses

- On l. 410 the authors call the phylogenetic framework "robust," which implies that the tree is generally suitable for testing dN/dS for all genes. However, Figure S2 clearly shows substantial gene tree discordance, so I am unclear why the framework is "robust" for gene-by-gene testing.

This is an interesting observation and we agree with the reviewer on the downsides of ignoring the gene tree heterogeneity. However, we think that the alternative of assuming the topology derived from all loci is a more conservative approach to testing dN/dS than using individual single-gene trees. In this context, we use the ML tree for two purposes: (i) reconstruct ancestral character states of ecological and life-history traits for a conservative definition of taxa partitions, and (ii) calculate dN/dS in PAML. We argue using the ML tree is more conservative than using single gene trees, which given stochastic errors in gene tree inference could artificially inflate not only dN/dS values but also the number of transitions in ancestral character state reconstructions. The downside is that gene tree heterogeneity is ignored in our analyses, but we think our approach is more conservative. In any case, repeating the PAML analyses with gene trees did not substantially change the obtained patterns, as it is mentioned in the manuscript.

- On l. 416 what does "markedly different" mean specifically? Are the distributions statistically different? They seem to overlap with each other and the background distribution.

The reviewer is right and “markedly” has been removed.

- While the targeted sequencing of putatively "adaptive" loci was an effective way to increase sampling depth, I am unsure how to interpret these results or draw general conclusions about the

entire genome. It seems more than plausible that other loci might have been under even stronger selection that were simply not sequenced, so I would recommend recontextualizing the results in the last section about diet to note that this result is only reported within the context of the pre-selected, targeted genes sequenced here and do not represent a naive genome-wide or random RAD-set of loci.

The reviewer is totally right. We have added the following sentences in the referred section: “The molecular evolutionary analyses of the of 29 selected innovation-associated loci revealed interesting patterns associated with adaptive radiations, but also with ecology and life history. Nevertheless, the generality of such results will need to be confirmed, refuted, or adjusted using larger genome-wide approaches such as genome re-sequencing.”

- Also recommend being (even more) specific about what is meant by "neutral"? As in "not under recent selection post-radiation", etc.

As mentioned earlier, we agree that “neutral” is not appropriate, so it has been removed from the text.

Minor Points:

- I realize it is still conventional (unfortunately, in my opinion) to use bootstrap/PP the branch score annotations for the primary figures and relegate IC scores to the supplement. However, the problems (and lack of information) with bootstrap/PP for large-scale data have been demonstrable in nearly every recent phylogenomic study of the last 5-10 years. I would encourage the authors to consider at least including the IC scores alongside the bootstrap/PP in Figure 1 or even swapping the scores used in figures 1 and supp fig 2. To me supp. figure 2 contains far more information that is useful than the presented Fig 1 tree that is almost completely perfect scores (when in fact many of the sub-familial branches have quite low IC scores). This argument is made verbally, but I think it would be impactful to show this in graphical form in the main paper. Depending on one's view of IC scores or the limitations of bootstrap in large datasets, this may be viewed as a rational or radical suggestion. Therefore, I am putting it in the "minor points" and I leave it to the authors' and editor's judgment.

We agree with the reviewer and have visualized IC values onto Fig. 1 by colouring branches in grey scale. The reader is referred to Supplementary Fig. 2. for detailed values of IC and ICA.

I. 114 "modeling"

II. 169, 332 and 352: "Patterson" misspelled and "D" should be italicized or math-type

I. 204: "basal" here is not necessary (and misused), recommend rephrasing to "B+B share a common ancestor distinct from other African..."

I. 210: "also" is redundant

I.404: "targeted"

I. 417: "toorh"

Changed as suggested.

I. 407: What does "neutral" mean here?

This sentence has been rephrased (see above) and “neutral” is not used anymore.

- [CH Martin et al 2015 Evolution] seems like a notably absent citation as one of the most comprehensive refutations of Cichlid sympatric speciation.

We thank the reviewer for pointing out this relevant paper, which is now referred to in the revised manuscript.

Reviewer #2:

The authors used a targeted capture and sequencing of about 1 Mb of aligned sequence from 149 species of Tanganyika and related worldwide cichlids with only 9% missing data. They use this data for 1. constructing a well-resolved species phylogeny, 2. time the phylogenetic events, 3. infer basal hybridisation events by D statistics, 4. survey natural selection on ecologically relevant genes in the targeted regions. They conclude 1. that phylogenetic timings fit biogeography very well, 2. that hybridisation occurred early and likely provided important genetic variability for diversification, and 3. that positive selection has been prevalent on opsin genes. I agree with 1. and 3. but am worried about 2., see below. In general, it is a well-designed study, competently carried out and well written and understandable even for a non-cichlid expert as myself. I think that the results are important for our understanding of the cichlid diversification globally and in Tanganyika in particular as a perhaps "easier" situation to solve than the later Malawi/Victoria radiations. The illustrations are generally nice, though a bit cramped. It would help a non specialist if geographical location was marked for each species in Figure 2 and Figure 4 has to be made larger to be readable.

We are glad the reviewer found our work of interest, well designed and carried out, and thank her/him all the thoughtful comments. Regarding the figures, we added current distribution patterns to Fig. 2 and improved readability of Fig. 4.

I have one important concern: As I understand it sequence reads are trimmed and aligned to a lake Malawi cichlid reference (Maylandia Zebra) and the consensus sequence is then called for each species. This might easily cause reference biases and such biases will depend both on the polymorphism level within each individual tested and its phylogenetic distance to the reference (which unfortunately is not the same for all individuals since the reference is an ingroup). When D statistics are used in e.g. primate research, it is common to use mapping to a species that is an outgroup to the set of species tested. I fear that species closer to the reference have a larger chance of being artificially closer to the reference by this approach because it is likely that the reference allele will be called in the consensus whenever the investigated individual is heterozygous (because such reads will map better). D statistics are VERY sensitive to even the smallest biases so my fear is that some of the hybridisations discussed might be artefacts. Perhaps it is not the most likely, but given the importance of such findings, we need to be surer. I suggest that the authors do at least the following two sets of analyses

1. Use the same reference as before but calling all the polymorphisms within individuals instead of a consensus. Then D statistics can be rerun but now using a random haplotype from the diploid genotype. I think results will change but do not know how much. It would also be great to have measures of heterozygosity for each individual, and this could be used to see whether shared polymorphisms are prevalent and how far back. This could also feed positively into the dn/ds analysis that would be sensitive to a too high level of shared polymorphisms
2. Use a different reference from a different part of the phylogeny and repeat the complete analysis and show that the D statistics are robust to this. If available, an outgroup reference would be preferable, but if not, it should be a reference which is in a different part of the phylogeny with respect to the hybridisations inferred.

The reviewer is totally right to point out the possibility of Maylandia zebra being a problematic reference for read mapping. However, this was a misunderstanding because our description of the methods was not appropriate. We have improved the description of the mapping pipeline to make it clear that the pipeline uses a quasi-de novo assembler and thus the choice of a specific reference will have no the suspected effect. The pipeline is described in detail in Hamilton, et al. 2016. (BMC Evol. Biol. 16:212), to which we also refer in the text.

The procedure for assembling loci and calling consensus alleles was as follows: it uses a diverse set of references as a starting point to map reads to each locus, provided that reads have a similarity >55% to any one of the references for a 100bp stretch. Because of this low similarity threshold, the initial

mapping step is very insensitive to the set of references employed. Once a few reads are mapped for a particular locus, those reads act as reference for a more refined mapping of the remaining reads, which are then assembled de-novo for each locus. In general, any allele from any homolog with 55% similarity will be assembled into a locus, with sequences diverging less than 5% from each other being clustered as alternative alleles of the same locus. For each locus, consensus sequences are called based on the probability of producing the pattern of observed bases at each site having resulted from a sequencing error compared to two heterozygous alleles and ambiguity is called if polymorphism is more likely derived by heterozygous allelic state at that site.

Given the quasi-de novo approach of our pipeline, our consensus sequences should be robust to the set of references used for the initial relaxed mapping and thus repeating analyses with different sets of references would not make a difference.

Following suggestion #1, we re-computed all D-statistics using phased alleles (one randomly chosen haplotype). The new results show more robust patterns than previous ones based on consensus sequences. We also tested the two main introgression hypotheses in Figure 3 with different outgroup taxa to rule out the possibility of outgroup biasing the D statistic. This addresses suggestion #2. We thank the reviewer for this suggestion, which has allowed us to critically revisit our methods and increase the confidence in the new results.

Reviewer #3:

The authors aim to understand the evolutionary history relating to the deep structure of the Lake Tanganyika radiation of cichlids, given the poorly understood interrelationships among tribes. The goal is to understand, in particular, the origin and evolution of key innovations and the role of hybridization in shaping the adaptive radiations of the cichlids. To this end, they use an anchored hybrid enrichment approach on the Lake Tanganyika species flock, generating a robust time-calibrated tree of cichlids for use as framework to study patterns of molecular evolution.

The premise for the paper is intriguing and important. The authors lay out the arguments clearly, and explain the very obvious need for the work and the rationale for their approach. I thoroughly enjoyed reading the manuscript and see it as an important and very valuable contribution.

We are pleased the reviewer found our research relevant and thank her/him for all the useful suggestions.

For improving the manuscript, there were two general concerns that came up:

1. Context. The study focuses entirely on cichlids. Starting from the first line of the introduction, almost every sentence in the text is related to cichlids and cichlids alone. While cichlids might be charismatic, the importance of such studies must still be placed within a bigger picture context. In what other lineages has adaptive diversification occurred in a similar way? In what others has adaptive admixture played a role? Why is this of general importance beyond cichlids?

Following the reviewer's suggestion, we have slightly shifted the focus of our manuscript (see below), and framed our findings into the bigger picture of understanding adaptive radiations. In the introduction we contextualize adaptive radiations and their importance in understanding biodiversity and speciation. Our results, particularly those about hybridization are compared to other adaptive radiations in cichlids but also other iconic radiations in animals and plants.

2. Hypothetical framework: It would help to have some hypotheses that could be tested. I think that this would make the arguments stronger. At the moment, the study is set up to “demonstrate the power of the anchored hybrid enrichment strategy”, which makes it seem like a Methods paper. It could be set up in the context of “We set out to address xx hypotheses in the context of the adaptive radiation of Lake Tanganyika cichlids: (1) timing will coincide with xxxx; (2) hybridization between xxxx might be expected based on findings in Lake Victoria cichlids; alternatively ... (3) selection tends to act most strongly on” etc. Such a framework would be straightforward, and the authors already have hypotheses for some elements, though buried in the Results (eg, lines 264-266 outlines alternative scenarios for timing; lines 377-390 start to talk about possibilities for hybridization scenarios based on previous studies; and lines 471-474 give the beginnings of hypotheses for the role of selection on opsin genes).

We are grateful for this recommendation, which we believe has substantially increased the interest and readability of our manuscript. We have emphasized our biological hypotheses and framed our work in the broader context of reconstructing the evolutionary relationships in an adaptive radiation and understanding the role of hybridization and the innovations using the Lake Tanganyika radiation as example.

Minor comments:

Line 151 “approaches now permit to generate large” Fix (“approaches now permit the generation of large”?)

Line 158 “has demonstrated to enhance “ Fix

Line 404 “Anchored sequencing of tragetted genes” Fix

Changed as suggested.

Lines 531-533. This last sentence makes the paper come off as methods focused. The results are really interesting, and it would make the work stronger to focus on what the methods have shown rather than the methods themselves.

As mentioned above, we have shifted slightly the focus our paper and thus, replaced this sentence with a more general statement: “Taken together, our findings contribute towards better understanding the patterns and processes shaping the great radiation of cichlids in Lake Tanganyika and East African lakes.”

Figure 2. is not at all clear. If the vertical blue shadows represent splits of Gondwanan-derived landmasses and the formation of Lake Tanganyika, which one represents what? The superimposed maps don't help much. It would help to mark on the tree the colonization of the lakes.

Following the reviewer's suggestion, we modified Fig. 2. We used different colours for shadows denoting Gondwana fragmentation and the Lake Tanganyika colonization, simplified the maps and indicated main distribution patterns and the events of colonization of African Great Lakes.

Reviewer #4:

The manuscript presents a study of the inter-tribal hybridizations in cichlids using target sequencing to obtain a large number of loci. The selective pressure on some of these loci is also investigated to characterize the adaptive evolution in the cichlids.

The manuscript is well written, but the text could be made more concise to help the reader see the main aspects of the results. At the same time, some analyses are lacking details, which leads to potential issues with these analyses.

First, the manuscript is quite lengthy and the authors should attempt to propose a more concise text. It will help the reader to understand the key elements that are of interest for a wider audience without being lost in the details. For instance, the introduction is long and the part on the timescale of cichlids evolution could be merged with the key innovation part and shortened. Similarly, the first part of the results and discussion on the anchored phylogenomics should be shortened as it is not the most interesting for non specialists in cichlids.

We thank the reviewer for all the thoughtful comments. In this revised version, we have significantly shortened and streamlined the text (particularly in the mentioned sections), while briefly expanding some sections as required by the reviewers' suggestions.

Second, the study hypothesize that hybridization events and the presence of ILS are affecting the evolutionary history of cichlids. It is thus surprising that a standard concatenation analysis returns the same well supported phylogenetic tree as a coalescent tree (lines 181). This would suggest that the evolutionary history was not affected that much by these events. I would like the authors to comment and discuss this aspect.

This is a very interesting observation. We have performed loci resampling analyses in order to understand this effect. We found that coalescence shows a better performance than concatenation for middle-size datasets (100 to 400 loci) but a similar performance with 500 or more loci, which suggests that with 500 or more loci the "genuine" phylogenetic signal prevails over the confounding effect of ILS in concatenation analyses (which do not account for ILS) and therefore the high concordance found when using the full dataset of 533 loci.

Third, the divergence time analyses are really far from being optimal. I can understand that with the large amount of data presented here, some complex analyses using BEAST might not be very practical, but the lack of convergence in the analyses are probably more due to the large heterogeneity in the rate of evolution present in the data than anything else. Indeed, it is evident from figure 2 that including the basal lineages in the analyses drastically increases the range of evolutionary rates, which, combined with the short rates in the cichlids, will then be difficult to model using a single distribution. Using state-of-the-art method is essential to have reliable estimates and to make sure that the different dates obtained compared to Matschiner et al are due to the larger data rather than methodological issues. It would be thus important to reduce the data to focus on the main cichlid tribes and thus reduce the heterogeneity in rates that is causing the lack of convergence.

As mentioned in the manuscript, a full Bayesian timetree analysis with our dataset was impractical, failing to converge after weeks of computation in high-performance GPUs. Therefore, we used RelTime, which has been shown to produce very similar estimates to Bayesian analyses in several empirical datasets (e.g., Mello et al. 2017 Mol Biol Evol 34:45-50). Because clock models implemented in both BEAST and RelTime are able to accommodate among-lineage rate variation, we think that the cause for the lack of convergence is unlikely the rate differences between the Lake Tanganyika flock vs. the outgroup. In fact, RelTime computes branch-specific rates (analogously to uncorrelated lognormal relaxed clocks do in BEAST), so branch rates do not follow a single distribution. In addition, excluding non-Tanganyikan outgroup is not possible, since they are essential

in order to date both the origin of the Lake Tanganyika flock and the deepest divergences in the tree. In addition, the computational efficiency of RelTime allows to run many independent analyses to obtain a good understanding of the effect of calibrations points (the most influential choices in molecular dating), such as the effect of the newly discovered *T. pickfordii* fossil or the misplacement of *Mahengechromis*.

In any case, following the reviewer's suggestion, we run several BEAST analyses after heavily reducing the number of taxa and genes in different ways (selecting the most informative loci by PhyInformR, those with least missing data or random loci), used different tree priors (Yule, birth-death) and clock models (uncorrelated, autocorrelated), and contacted BEAST developers for further help. But despite all our efforts, BEAST runs failed to reach convergence after several weeks of computation (200 Mio. generations).

Despite the lack of convergence, the divergence times obtained by different BEAST analyses on smaller datasets showed similar dates to those obtained with RelTime, which made us confident of the results presented in the paper. For example, a BEAST analysis using a reduced dataset (46 taxa and the 50 most informative loci), calibration scheme C10 with uncorrelated lognormal clock and birth-death tree prior run for 200 Mio. generations estimated very similar dates for the deeper nodes in the tree, dating the origin of cichlids at 129.6 Ma (vs. 128.4 Ma in RelTime) and the split between African and Neotropical cichlids at 95.9 Ma (vs. 101 Ma.) in agreement with Godwana fragmentation. The estimated age for the Lake Tanganyika flock was at 26.3 Ma (vs. 13.7 in RelTime), which is appreciably older but still closer to our estimate than the ~50 Ma. obtained by Matschiner et al. Despite roughly similar ages, it is difficult to compare the RelTime estimates based on the full dataset and a BEAST analysis based on a heavily reduced dataset. Despite differences in dating methodology and implementation of different models (not directly comparable between both software), the selection of taxa and loci surely is at play too. The selection of loci in reduced dataset would raise questions about its representation of the full dataset, and removing taxa could produce biased estimates due to the underestimation of branch lengths (see Bromham et al. 2018 Biol. Rev. in press 10.1111/brv.12390).

Nevertheless, we totally agree with the reviewer in the necessity of clarifying the apparent incongruence between existing cichlid molecular datings, including that of Matschiner et al. A comprehensive understanding of the causes behind this apparent conflict of timetrees requires an in-depth analysis comparing several genome-scale datasets, different dating software and comparable parameterizations (clock models, calibration settings, etc.). In addition to the problems highlighted above, we feel that all the analyses and deep methodological discussion required by this complex topic would take too much space in the current paper and are out of scope. Instead, a better way of addressing the question is to devote an entire separate methodological paper where all the relevant points can be carefully addressed. In fact, and moved by the reviewer suggestion, we have established a collaborative effort with the group of Walter Salzburger (University of Basel) and Michael Matschiner (University of Oslo) for a meta-analysis paper that addresses all the points mentioned above.

In summary, we think the molecular dating analyses presented here can stand on their own and aim for a deeper understanding of this controversy in a future study. To stress this point, the following sentence was added in the manuscript: "Further insight into the apparent contradiction between existing cichlid timetrees would be possible through conducting an in-depth study comparing different dating software/methodologies and molecular datasets with comparable assumptions and models".

You should also clarify how the uncertainty in the calibration points were incorporated in the analyses (248-249). This is critical especially as the dates obtained differ from some in the literature.

According to the RelTime implementation, calibration points were incorporated using the minimum and maximum bounds, e.g. similar to assuming uniform probability, which is preferable when the probability distribution of calibrations are not known precisely (Tamura et al. 2018 Theoretical

foundation of the RelTime method for estimating divergence times from variable evolutionary rates. bioRxiv <http://dx.doi.org/10.1101/180182>). This has been clarified in the methods.

Fourth, the criteria used to select the set of genes for the target sequencing are unclear. The authors should provide a clear description of the procedure used and which gene regions were targeted and why. You further say once that you have 260 + 29 gene regions (line 555-560) while you talk about "533 aligned anchored loci" (line 177). Could you clarify how these two numbers were obtained?

Anchored loci were designed for their good phylogenetic performance and low-copy number, and this has now been specified in the text. The 29 candidate genes were selected based on previous literature and it is also clarified in the text (see also response to Reviewer #1). Anchored loci came from 260 gene families, and generally speaking each anchored locus corresponds to one of the exons plus the flanking non-coding regions. In the same way, the 29 candidate genes correspond to 174 loci. This has been now clarified in the main text.

Fifth, I do not understand the arguments given in lines 260-266. Why do you argue that an initial divergence in the river system would lead to tribes having broader niches? This is entirely dependent on the type of ecological niches that you will have in the different river system. You could have very similar niches in these river systems, which would lead to narrow niches for the different tribes. In any case, I am not sure that you have the data to show this and this argument does not really hold.

As noted in the text, the necessarily broad confidence intervals of timetrees do not allow us to rule out whether the initial divergences between Lake Tanganyika cichlids occurred together or before the lake formation. The referred lines express our best attempt to congruently explain molecular dating, ecology and distribution of Tanganyikan endemic species, but is it necessarily an hypothesis and thus we have toned down this section and clarified the assumptions and implications for both scenarios. We argue that a radiation coincident with the lake formation is more likely given that all Tanganyika tribes (except for the early-branching Lamprologini) occupy distinct niches supports our hypothesis. This of course assumes that lakes can provide more diverse niches than rivers. Additional support for an intra-lacustrine radiation is the general lack of riverine representatives from most Lake Tanganyika cichlid tribes, as under the alternative scenario we would expect to find riverine members among Lake Tanganyika tribes as happens in closely related non-Tanganyika tribes. These points have been discussed further in the text.

Sixth, the hybridization and ILS events are investigated using Patterson's D, but it is unclear how the authors set up the topological constraints to perform these tests. It would be good to have more details about the expectations tested.

We have now clarified this aspect into the methods: "For each hypothesis, relevant groups were selected following in the ML topology (Supplementary Fig. S1) to create asymmetric trees of four taxa, as required by Patterson's test, that allowed detecting all hypothesized gene flow events", as well as in the caption of Fig. 3 (see also response to Reviewer #1).

On a similar note, the groups tested include different species, and not population level data as in the standard test. How did you deal with the potential substitutions occurring within the groups formed? Did you remove these sites before applying the D test?

The reviewer is right. Substitutions occurring within each of the groups could confound population-level analyses of the D test, which pool together different species belonging to a particular tribe of lineage being tested. For this reason, population-level D tests have been removed from the revised manuscript (see also response to Reviewer #1). In contrast, individual-level D tests, which iteratively test all possible combinations of 4 individuals one belonging to each of the 4 groups, do not suffer from within-group substitutions. The distribution of D values provided by individual-level analyses of D allows better understanding the events of gene flow between groups.

Finally, I am a bit puzzled by the selection test done and more specifically the significance level used. You tested 29 genes and each gene has a certain probability to be significant. You are thus clearly in a multiple testing problem and the threshold for the p-value should be adjusted (see for example Studer and Robinson-Rechavi, 2008).

Following the reviewer suggestion, now we account for multiple testing using a Benjamini-Hochberg False Discovery Procedure to adjust significance of the individual tests.

Further, the M2a/M1a test assumes that the sites are under selection across the whole phylogenetic tree, which in your case means that it is also under selection in the basal lineages of the phylogenetic tree. Did you include these basal lineages in the analyses and, if yes, how can you interpret these results as adaptation interesting for the Lake Tanganika cichlids? The clade model will give you this information, but not the site model.

As correctly pointed out by the reviewer, these site models average dN/dS across branches in the tree, and thus cannot be used to look for adaptations in Lake Tanganyika cichlids if also the outgroup is included. In fact, we use site models to (i) obtain general dN/dS values per gene in order to compare the three gene categories (Fig. 4a) and (ii) identify the genes showing evidences of positive selection, which are further studied using clade model C. All our claims of genes that are potentially adaptive are based on Clade model C analyses, which account for among-lineage heterogeneity between the defined partitions, and never on the previous site models.

You also argue that ecological traits explain sequence variation better than null models (line 491). How did you do these analyses exactly? Did you run the clade test on several partition of the data and took the one that best explained the evolution of the gene? How exactly and did you correct for multiple testing? This should be clarified and the details of the analyses given as it is a key result of the manuscript.

For each gene, several CmC were run, which partition the data by lineages or ecology/ life history traits, and then compared by AIC (as specified in the methods). To clarify this further, we now explicitly show differences between AIC values in the main text in support of our claims. The full details about CmC models including AICs are available in Supplementary Tables 40-44. Additionally, likelihood ratio tests between a specific model and the null model (i.e., M2a-rel) were conducted and presented in the Supplementary Tables, and the significance level for such test was adjusted for multiple testing using a Benjamini-Hochberg False Discovery Procedure (as described above).

Minor comments

- line 78: change to "pertains to the deep"

Changed as suggested.

- line 102: it is well known that even with full genomic data, we will not be able to estimate the true species tree without some errors. Using a cloud of trees will then always be necessary.

As correctly pointed out by the reviewer, every phylogenetic inference is, by definition, an approximation of the true evolutionary history. The galled network in Supplementary Fig. 4 provides interesting information about the gene tree conflict present in the data and we believe the new Fig. 1 provides a visual assessment of this conflict similarly to a "cloudogram" while allowing us to also still see the obtained phylogenetic hypothesis.

REVIEWERS' COMMENTS:

Reviewer #1 (Remarks to the Author):

This revision by Irisarri et al is thoughtful and comprehensive, and I found the new framing of the study greatly improved the clarity and specificity of the findings.

The interweaving of IC and bootstrap, ASTRAL and RAXML gives the description of a tree a very current and modern feel in describing the ILS patterns and what parts of the tree are solid and which are a little more tenuous.

The use of the D-statistics seems overall much more conservative and focuses on clear and significant introgression trends that are informative.

The shift to focusing on candidate loci involved in specific functional traits is a really nice reframing and overall the flow and language of the manuscript is much improved (even from an originally strong manuscript). The functional site analysis is carefully constructed and is reported in a way that communicates the known inherent limitations of the PAML site-based analyses.

The length of the manuscript and its organization are sound and the figures are generally clear and attractive (see a few notes below). The comprehensiveness of the supplements and methods is appreciated, and all aspects of the study seem replicable given the level of detail provided.

Overall, I think this helps clarify and summarize the Cichlid radiation in a comprehensive manner, and specifically in a manner focuses on contextualizing and synthesizing the major trends rather than overextending into too many sub-hypotheses. I recommend acceptance of this manuscript, and have only some minor notes about the figures below.

Minor notes:

For Figure 3B and 3C, it might be simpler just to use short abbreviations that look like the names to the left (i.e., Lake Tanganyika = LT, Tilapiini = TI, etc.) instead of using the P1,P2. While this is conventional in D-stats, it means you have to do an extra step of cross-referencing the P* labels in order to interpret the figure.

The subpanel labels (a-c) on Figures 3 and 4 are completely different, and probably Fig 4 are too large.

l.58: "silverswords" is misspelled

Supplementary Table 13: the axes on the D-stat are completely illegible, but otherwise this is clear and comprehensive.

Reviewer #2 (Remarks to the Author):

The authors have done a nice job addressing the comments from me and the other reviewer. There are still issues that could be discussed in terms of interpretation, but I believe this is for future studies to settle so I am happy to let the authors go with their current interpretation since I now feel convinced that the science is well carried out and the results should appeal to a broad audience

Reviewer #4 (Remarks to the Author):

The revised manuscript considered all the comments given by the reviewers and modified the text of the manuscript accordingly. The authors did a good job in replying to the comments and this improved the manuscript. I have a couple of comments that are remaining at this stage.

First, I am coming again with the interpretation of the concatenated vs coalescence based estimation of the species tree. I understand your reply and it does seem that there is enough signal to get a species tree (lines 164-167), but, under ILS, increasing the number of loci should not decrease the incongruence because ILS is a random process affecting each locus. The pattern that you get would look more similar to the presence of gene flow that might be affecting differently the loci sequenced. I think that you should be more careful in the interpretation that you give in the current text.

Second, and on the same subject, I find the sentence on line 191-192 very awkward. It is true that ASTRAL accounts for ILS, but how can you then argue that the difference is due to hybridization instead of ILS (line 191 seems to make this causal link, although the line 280 has a better formulation). As said in my previous comment, the two processes are not the similar result obtained with ASTRAL and RAxML is in my opinion not due to the fact that ASTRAL account for ILS but because you have hybridization occurring.

Third, on line 381, it is very unclear what is the analysis underlying the rate changes shown here. Is it the clade model, in which case, this should be explained clearly. If so, please also give the omega values obtained to have a better idea of the rate difference.

Minor comments:

- line 373: please give the range of omega values and/or p-value in the text, it would help the reader to see the selective forces present.
- I would also add a table in the text to summarize the calibration constrains used. Even if it is explained in the method section, it would help the reader to better see what is different between the C01 to C10 scenarios.

Response to Reviewers' comments

Reviewer #1

This revision by Irisarri et al is thoughtful and comprehensive, and I found the new framing of the study greatly improved the clarity and specificity of the findings.

The interweaving of IC and bootstrap, ASTRAL and RAxML gives the description of a tree a very current and modern feel in describing the ILS patterns and what parts of the tree are solid and which are a little more tenuous.

The use of the D-statistics seems overall much more conservative and focuses on clear and significant introgression trends that are informative.

The shift to focusing on candidate loci involved in specific functional traits is a really nice reframing and overall the flow and language of the manuscript is much improved (even from an originally strong manuscript). The functional site analysis is carefully constructed and is reported in a way that communicates the known inherent limitations of the PAML site-based analyses.

The length of the manuscript and its organization are sound and the figures are generally clear and attractive (see a few notes below). The comprehensiveness of the supplements and methods is appreciated, and all aspects of the study seem replicable given the level of detail provided.

Overall, I think this helps clarify and summarize the Cichlid radiation in a comprehensive manner, and specifically in a manner focuses on contextualizing and synthesizing the major trends rather than overextending into too many sub-hypotheses. I recommend acceptance of this manuscript, and have only some minor notes about the figures below.

We are glad that the reviewer found our new revised manuscript significantly improved, after we addressed the reviewers' previous concerns and suggestions.

Minor notes:

For Figure 3B and 3C, it might be simpler just to use short abbreviations that look like the names to the left (i.e., Lake Tanganyika = LT, Tilapiini = TI, etc.) instead of using the P1,P2. While this is conventional in D-stats, it means you have to do an extra step of cross-referencing the P* labels in order to interpret the figure.

The subpanel labels (a-c) on Figures 3 and 4 are completely different, and probably Fig 4 are too large.

Changed as suggested.

l.58: "silverswords" is misspelled

Changed as suggested.

Supplementary Table 13: the axes on the D-stat are completely illegible, but otherwise this is clear and comprehensive.

Violin plots have been modified to allow better visualization of the plot axes.

Reviewer #2

The authors have done a nice job addressing the comments from me and the other reviewer. There are still issues that could be discussed in terms of interpretation, but I believe this is for future studies to settle so I am happy to let the authors go with their current interpretation since I now feel convinced that the science is well carried out and the results should appeal to a broad audience

We are pleased that the reviewer found her/his concerns satisfactorily addressed in the revised version, and the underlying science to be carried appropriately.

Reviewer #4

The revised manuscript considered all the comments given by the reviewers and modified the text of the manuscript accordingly. The authors did a good job in replying to the comments and this improved the manuscript. I have a couple of comments that are remaining at this stage.

We were pleased to find reviewer #4 also found our revisions correctly addressed her/his concerns, leading to an improved manuscript.

First, I am coming again with the interpretation of the concatenated vs coalescence based estimation of the species tree. I understand your reply and it does seem that there is enough signal to get a species tree (lines 164-167), but, under ILS, increasing the number of loci should not decrease the incongruence because ILS is a random process affecting each locus. The pattern that you get would look more similar to the presence of gene flow that might be affecting differently the loci sequenced. I think that you should be more careful in the interpretation that you give in the current text.

The reviewer is right. The observed increased resolution power of larger datasets do not only reflect an increased phylogenetic signal, but can also be the cause of introgressed loci having an overall smaller misleading effect in larger datasets. Therefore, we have modified this section as follows:

“Using 500 or more loci instead, both methods converge to very similar topologies, suggesting that the genuine phylogenetic signal can overcome the confounding effect of ILS ignored by concatenation. In addition, using more genes probably dilutes the adverse effect of gene flow events (e.g. introgression) on both concatenation and coalescence approaches.”

Second, and on the same subject, I find the sentence on line 191-192 very awkward. It is true that ASTRAL accounts for ILS, but how can you then argue that the difference is due to hybridization instead of ILS (line 191 seems to make this causal link, although the line 280 has a better formulation). As said in my previous comment, the two processes are not the similar result obtained with ASTRAL and RAxML is in my opinion not due to the fact that ASTRAL account for ILS but because you have hybridization occurring.

We agree that this sentence is misleading, so we have rephrased it and simply say: “We interpret this difference to result from hybridization (see below)”.

Third, on line 381, it is very unclear what is the analysis underlying the rate changes shown here. Is it the clade model, in which case, this should be explained clearly. If so, please also give the omega values obtained to have a better idea of the rate difference.

Yes, these results are based on Clade model C, which is not reported together with the three omega (dN/dS) values for the three groups involved.

Minor comments:

- line 373: please give the range of omega values and/or p-value in the text, it would help the reader to see the selective forces present.

Following the reviewer's suggestion, we have now included relevant omega (dN/dS) values in the main text.

- I would also add a table in the text to summarize the calibration constrains used. Even if it is explained in the method section, it would help the reader to better see what is different between the C01 to C10 scenarios.

Such a table is present in the supplement (Supplementary Table 2). As already mentioned by the reviewer, the most important differences between calibration schemes are already highlighted in the main text in order to facilitate the interpretation of our timetree analyses. For additional details, including all the references, the reader is directed to Supplementary Table 2.